# ProteinAE: Protein Diffusion Autoencoders for Structure Encoding

**Shaoning Li**[1,2]*, **Le Zhuo**[1,3]*, **Yusong Wang**[2,4], **Mingyu Li**[5], **Xinheng He**[6], **Fandi Wu**[7], **Hongsheng Li**[1], **Pheng-Ann Heng**[1]

[1] CUHK [2] ZGC Academy [3] Krea AI [4] XJTU [5] SJTU [6] AstraZeneca [7] Tencent

## Abstract

Developing effective representations of protein structures is essential for advancing protein science, particularly for protein generative modeling. Current approaches often grapple with the complexities of the $SE(3)$ manifold, rely on discrete tokenization, or the need for multiple training objectives, all of which can hinder the model optimization and generalization. We introduce ProteinAE, a novel and streamlined protein diffusion autoencoder designed to overcome these challenges by directly mapping protein backbone coordinates from $E(3)$ into a continuous, compact latent space. ProteinAE employs a non-equivariant Diffusion Transformer with a bottleneck design for efficient compression and is trained end-to-end with a single flow matching objective, substantially simplifying the optimization pipeline. We demonstrate that ProteinAE achieves state-of-the-art reconstruction quality, outperforming existing autoencoders. The resulting latent space serves as a powerful foundation for a latent diffusion model that bypasses the need for explicit equivariance. This enables efficient, high-quality structure generation that is competitive with leading structure-based approaches and significantly outperforms prior latent-based methods. Code is available at https://github.com/OnlyLoveKFC/ProteinAE_v1.

## 1 Introduction

Proteins serve as the foundation of life. Protein structures, determined by the folding of various amino acid sequences, dictate their biological functions and behaviors. Understanding and representing these structures is important for various protein tasks, typically protein generative modeling. A dominant paradigm for visual generative models involves a two-step process (Esser et al., 2021; Rombach et al., 2022): initially compressing the pixel input into a compact latent representation with visual autoencoders (tokenizers), and then performing generative modeling within this latent space. This paradigm significantly enhances both efficiency and performance of modeling complex visual distributions. In contrast to the visual domain, which are typically represented at the pixel level and are high-dimensional, protein structures are represented by continuous atom coordinates in a lower dimensional space, i.e., 3D Euclidean space ($E(3)$) and differ greatly in size due to their diverse amino acid sequence lengths (Jumper et al., 2021; Abramson et al., 2024). These characteristics require new recipes for creating protein structure autoencoders and necessitate the exploration of more efficient protein generative modeling methods.

Researchers have made several attempts to encode protein structures with autoencoders. Pioneering efforts in this area include the ESM3 VQ-VAE tokenizer (Hayes et al., 2025) and the DPLM-2 lookup-free quantization (LFQ) tokenizer (Wang et al., 2024; 2025), which were among the first to convert continuous 3D coordinates into discrete tokens to jointly model sequences and structures used in generative masked language models. (Yuan et al., 2025) proposes AminoAseed with further codebook improvements. While achieving initial success, these autoencoders are not ideal in the following aspects: (i) operating on the intricate $SE(3)$ manifold (involving translations and frame rotations) necessitates incorporating equivariance and physical constraints, thus introducing complexity to the latent space and the design of model architecture; (ii) discretizing continuous atom coordinates to tokens leads to a loss of reconstruction accuracy; (iii) requiring the combination of

---

*Equal contribution.

complex training objectives (e.g., FAPE loss, distance loss, violation loss, KL loss, etc.), which demands tuning individual weights for each term; (iv) being limited to a fixed input size (sequence length) and lacking a compact bottleneck latent space for efficient generative modeling. These raise the question: *Could we design a simpler, more accurate, and effective protein autoencoder in a continuous, compact latent space?*

In this work, we propose PROTEINAE, a protein diffusion autoencoder designed for effective and efficient structure encoding and generation. Specifically, PROTEINAE operates in a non-equivariant manner, and conducts autoencoding protein backbone atoms ($C_\alpha, N, C, O$) directly on E(3), avoiding the discretization. Inspired by recent works on denoising autoencoders (Chen et al., 2025), PROTEINAE employs a simple diffusion loss for training, which makes the learned structural representations maximize the ELBO of the likelihood of the input protein structures, which is similarly validated by AlphaFold3 (Abramson et al., 2024). An architecture comparison between the traditional protein autoencoder ESM3 structure autoencoder and PROTEINAE is shown in Fig. 1. ESM3 structure autoencoder employs a standard VQ-VAE framework (Esser et al., 2021) with a modified geometry-based transformer encoder on SE(3) and focuses more on the local structural pattern. In contrast, PROTEINAE's encoder and decoder are designed with scalable Diffusion Transformers (DiT) (Peebles & Xie, 2023). The encoder maps the

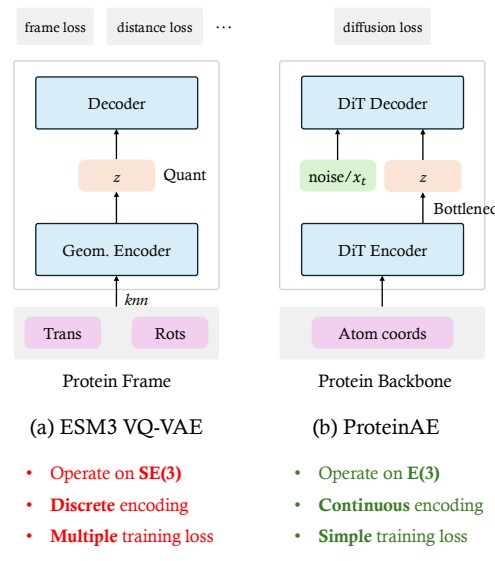

Figure 1: Comparison of ESM3 VQ-VAE and our PROTEINAE.

protein backbone atom coordinates to a latent representation $z$. We also incorporate a bottleneck design by downsampling the protein in length and channel dimensions, which is common in vision for efficiency (Ronneberger et al., 2015). The decoder takes $z$ as a condition and predicts the clean structure or velocity from noisy structures. At inference time, protein structures are reconstructed from the noisy coordinates with latent representation $z$ via a diffusion sampler.

Building on this protein autoencoder, we can further perform downstream structural latent analysis, such as protein latent diffusion modeling (PLDM) and physicochemical prediction. Previous generative methods are mostly structure-based diffusion models over SE(3) (Yim et al., 2023b;a; Watson et al., 2023; Bose et al., 2023), but they face challenges with equivariance and physical constraints such as bond lengths and angles (Yim et al., 2025). Protein multi-modal models like ESM3 use iterative decoding to generate discrete structure tokens, but their generation quality is not as expected. (Yim et al., 2025) proposes a hierarchical structure latent diffusion model, but the latent diffusion is conducted on the contact map and still relies on FrameFlow (Yim et al., 2023a) at the second fine-grain stage. Based on the lightweight PROTEINAE, we develop a scalable PLDM. It employs a standard DiT architecture, bypassing the need for explicit equivariance or physical constraints, and eliminates the computationally expensive triangle attention module for better efficiency.

We rigorously evaluate PROTEINAE on the CASP14 and CASP15 benchmarks, demonstrating state-of-the-art reconstruction quality. We further show that the learned latent space is highly effective for physicochemical property prediction. Finally, our PLDM is competitive with leading structure-based generative models and substantially outperforms existing latent-based methods in both sample quality and efficiency. Our primary contributions are:

- **PROTEINAE: A Simple and Effective Protein Diffusion Autoencoder.** We introduce a non-equivariant autoencoder based on Diffusion Transformers that operates directly on backbone atom coordinates in E(3). It learns a continuous, compact latent representation using a single flow-matching loss, avoiding the complexities of SE(3) manifold and discrete tokenization.

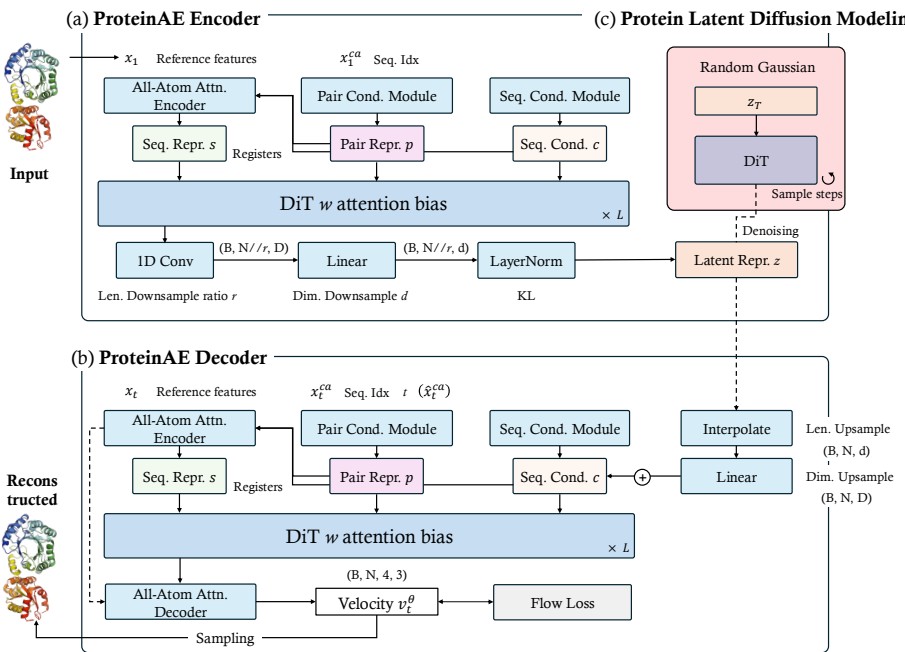

Figure 2: **Overall architecture of PROTEINAE.** (a) The encoder maps a protein structure to a latent representation $z$; (b) The flow decoder predicts the velocity field $v_t^\theta$ for structure reconstruction conditioned on $z$; (c) Downstream tasks like PLDM operating over the learned latent space.

- **High-Fidelity Encoding for Downstream Generative and Predictive Tasks.** PROTEINAE achieves state-of-the-art protein structure reconstruction. Its learned latent space enables accurate physicochemical property prediction and serves as the basis for protein latent diffusion model (PLDM) that significantly outperforms latent-based methods and is competitive with leading structure-based generative approaches with improved efficiency.

## 2 PROTEINAE: PROTEIN DIFFUSION AUTOENCODERS

As illustrated in Fig. 2, PROTEINAE employs an encoder-decoder architecture for learning latent protein structure representations. We explain abbreviations of some components in Fig. 2 in *italics*.

**Feature Preparation** Both the encoder and decoder take a protein backbone structure $x \in \mathbb{R}^{N \times 4 \times 3}$ as input ($x_1$ for the encoder, representing the clean structure, and $x_t$ for the decoder, representing the noisy structure at timestep $t$; $N$ denotes the protein length). Given the input structure, initial residue-level features are constructed. The pair representation $p$ (*Pair Repr.*) and sequence condition feature $c$ (*Seq. Cond.*) are generated through the Pair Condition Module (*Pair Cond. Module*) and Sequence Condition Module (*Seq. Cond. Module*), respectively. For the encoder, inputs to these modules include $x^{ca}$ and sequence index features. For the decoder (inputs shown in gray below), the diffusion timestep $t$ and self-condition $\hat{x}_t^{ca}$ are added to the Pair Condition Module inputs, and the latent representation $z$ is added to the output of the Sequence Condition Module. Reference amino acid features $\mathbf{f}$ are used by both modules. The feature preparation process can be summarized:

$$p = \text{PairCondModule}(x^{ca}, \mathbf{f}^{\text{seq\_idx}}, t, (\hat{x}_t^{c_a})), \tag{1}$$

$$c = \text{SeqCondModule}(\mathbf{f}) + z. \tag{2}$$

Note that inputs shown in gray are specific to the decoder. These initial token-level features ($p$ and $c$) are used alongside the atom-level input $x$. A lightweight All-Atom Attention Encoder (*All-Atom Attn. Encoder*) processes the atom-level features from $x$ to generate a token-level sequence

representation $s$ (*Seq. Repr.*). This module also outputs skip connection features $(q^{\text{skip}}, c^{\text{skip}}, p^{\text{skip}})$ used later in the decoder pipeline. The output of the All-Atom Attention Encoder is:

$$s, q^{\text{skip}}, c^{\text{skip}}, p^{\text{skip}} = \text{AllAtomAttnEncoder}(x, p, c, \mathbf{f}). \tag{3}$$

More details regarding the All-Atom Attention mechanism are provided in the next paragraph.

**Diffusion Transformers and All-Atom Attention**  Unlike many previous structure-based diffusion or representation learning models that rely on geometric equivariance, PROTEINAE adopts a non-equivariant architecture, forming the core of our feature processing, encoder, and decoder. This choice aligns with recent works such as AlphaFold3 (Abramson et al., 2024) and Proteina (Geffner et al., 2025), which feature stacks of conditioned and biased multi-head self-attention layers combined with transition blocks and residual connections. Operating on sequence and pair representations, these layers are optionally enhanced with attention biases derived from geometric relationships in the input structures. The input tokens are further augmented with additional registers (Darcet et al., 2023). To effectively handle variable-length protein inputs, we employ Rotary Positional Encodings (RoPE) (Su et al., 2024) instead of traditional absolute positional encodings. The Diffusion Transformer (DiT) architecture is utilized in three distinct parts of our model: (i) the All-Atom Attention modules during feature preparation and decoding, (ii) the core token-level Attention stacks in the PROTEINAE Encoder and Decoder, and (iii) the token-level Attention in the subsequent PLDM stage. We note that pair bias is incorporated in the DiT stacks of the PROTEINAE Encoder and Decoder, as well as within the All-Atom Attention modules, but it is omitted in the PLDM stage. A DiT block operating on sequence ($s$) and condition ($c$) representations, potentially incorporating pair bias ($p, \beta_{ij}$), can be summarized as:

$$s^l = \text{DiT\_pair\_bias}(s^{l-1}, p, c, \beta_{ij}), \tag{4}$$

$$s^l = s^l + \text{TransitionBlock}(s^l, c), \tag{5}$$

where $l$ denotes the $l$-th layer in a stack of $L$ blocks. $\beta_{ij}$ here represents the attention mask. A comprehensive algorithm is provided in the Appendix A.1.

To explicitly model and generate atom-level details for the backbone structure, we incorporate All-Atom Attention Encoder and Decoder modules, inspired by AlphaFold3 but implemented with fewer parameters. These modules utilize sequence-local atom attention, allowing interactions among all backbone atoms within a defined sequence neighborhood. This approach offers richer local interaction modeling compared to methods relying solely on KNN graphs, such as those adopted in the ESM3 VQ-VAE pipeline (see Fig. 1). The All-Atom Attention Encoder, described previously as part of feature preparation, aggregates atom-level features from the input structure into token-level representations. For consistency within the All-Atom Attention mechanisms, we fix reference features $\mathbf{f}$ (including positions, charge, mask, elements, and atom name characters) based on the standard Glycine (GLY) residue from the Chemical Components Dictionary (CCD). The All-Atom Attention Decoder operates after the DiT stack in the decoder pipeline. It takes the final sequence representation $s^L$ from the DiT stack and broadcasts token-level features back to atom-level. After computing atom attention, it projects these features to predict the noise velocity $v_t^\theta \in \mathbb{R}^{N \times 4 \times 3}$ in coordinate space. The output prediction of the All-Atom Attention Decoder is summarized as:

$$v_t^\theta = \text{AllAtomAttnDecoder}(s^L, q^{\text{skip}}, c^{\text{skip}}, p^{\text{skip}}, \beta_{ij}). \tag{6}$$

Note that while the DiT stack and All-Atom Attention modules are distinct components operating in sequence, their underlying attention mechanisms share principles of conditioned and biased self-attention.

**Autoencoder Bottleneck**  To obtain a compact latent representation $z$ and improve the efficiency of subsequent generative modeling, we introduce two types of bottleneck compression within the encoder pipeline (Fig. 2a): a *length* bottleneck and a *dimension* bottleneck. Following the stack of DiT layers, we obtain the sequence representation $s^L$. For the length bottleneck, we apply one or more 1D convolution layers with a kernel size of 3 and a stride of 2. This downsamples the protein length from $N$ to $N_{\text{down}} = N/r$, where $r$ is the total downsampling ratio. Subsequently, for the dimension bottleneck, a linear layer projects the sequence representation from its token dimension

$D$ to a reduced bottleneck dimension $d$. This compression process can be summarized as:

$$z = \underbrace{\text{LinearNoBias}}_{\text{Dimension downsample}} \left( \underbrace{\text{Conv1d}(\text{stride} = 2, \text{kernel} = 3) . (\text{transpose}(s^L))}_{\text{Length downsample}} \right). \tag{7}$$

Here, $\text{transpose}(\cdot)$ denotes the batch transpose, transforming a tensor of shape $(B, N, D)$ to $(B, D, N)$ for convolution. The resulting latent representation $z$ has shape $(B, N_{\text{down}}, d)$.

In the decoder pipeline, the latent representation $z$ needs to be upsampled and expanded to match the target protein length $N_{\text{target}}$ and channel size of conditioning features before being added (Eq. 2). This upsampling process involves expanding the dimension of $z$ and then interpolating its length:

$$z_{\text{upsampled}} = \text{transpose} \left( \text{Interpolate}(\text{transpose}(\underbrace{\text{LinearNoBias}}_{\text{Dimension upsample}}(z)), N^{\text{target}}) \right). \tag{8}$$

**Replace KL regularization with LayerNorm**  Unlike traditional VAEs that employ a KL regularization loss on the latent feature $z$, we apply $\text{LayerNorm}$ without learnable scales following DiTo (Chen et al., 2025). This operation is applied to the output of the bottleneck (Eq. 7) and yields the final latent representation $z$ used for both the PROTEINAE decoder and PLDM training. This approach eliminates the need for KL loss weight tuning and empirically demonstrates better reconstruction performance. Furthermore, by normalizing the latent space in this manner, we can directly train the PLDM on $z$ without requiring additional normalization within the diffusion process (Rombach et al., 2022). In conclusion, this design choice simplifies the overall training procedure for both PROTEINAE and PLDM.

**Autoencoder Pipelines**  Having introduced the core architectural components, we now detail their arrangement within the PROTEINAE encoder and decoder pipelines (Fig. 2).

*Encoder* (Fig. 2a) takes a native protein structure ($x_1$) as input. As described in "Feature preparation", this structure is used to derive initial sequence ($s$), pair ($p$), and condition ($c$) features. These features are processed through a stack of DiT blocks incorporating pair bias. The resulting representation then undergoes a bottleneck stage, which compresses both the length and dimension of the feature maps (as detailed in Autoencoder Bottleneck). Finally, the compressed latent representation $z$ is normalized via LayerNorm before being passed to the decoder or used for downstream tasks.

*Decoder* (Fig. 2b) operates within the flow matching framework, taking a noisy protein structure ($x_t$) at timestep $t$ as input. Similar to the encoder, $x_t$ is used for feature preparation to obtain initial $s, p, c$ features. A key distinction is that the latent representation $z$ (from the encoder during training, or sampled during inference) is upsampled and incorporated as conditioning information into the sequence condition feature $c$. These conditioned features ($s, p$, and the modified $c$) are then processed by a stack of DiT blocks with pair bias, mirroring the encoder's architecture. The output from the DiT stack is fed into an All-Atom Attention Decoder, which predicts the velocity vector $v_t^\theta$ required for structure reconstruction via ODE integration (Appendix A.2).

**Reconstruction Loss**  PROTEINAE is trained using a simple flow-matching loss, with no other auxiliary losses. The objective is to train the model to predict the velocity vector field $v_t^\theta$ at noisy structure $x_t$ and timestep $t$, conditioned on the latent representation $z$. The target velocity field in flow matching is defined as $v(t) = x_1 - x_0$. The input structures $x_1, x_t$, the noise $x_0$, and the predicted velocity $v_t^\theta$ are all represented as tensors in $\mathbb{R}^{4N \times 3}$. The PROTEINAE's training objective is:

$$\min_\theta \mathbb{E}_{x_1 \sim p_{\text{ds}}(x), x_0 \sim \mathcal{N}(0,I), t \sim p(t)} \left[ \frac{1}{4N} \left\| v_t^\theta(x_t, t, z) - (x_1 - x_0) \right\|_2^2 \right], \tag{9}$$

where $p_{\text{ds}}(x)$ is the data distribution, $p(t)$ is the timestep sampling distribution, and $v_t^\theta(x_t, t, z)$ is the velocity predicted by the decoder network. Following common practice in structure-based flow matching models, we use the $t$-sampling distribution $p(t) = 0.02\mathcal{U}(0,1) + 0.98\mathcal{B}(1.9, 1.0)$.

## 3 EXPERIMENT

### 3.1 EXPERIMENTAL SETUP

**Dataset and Model Configuration** We use the AFDB-FS (Jumper et al., 2021; Lin et al., 2024) dataset for training PROTEINAE and PLDM. This is a large-scale dataset of single-chain protein structures derived from the AlphaFold Protein Structure Database (AFDB) through sequential filtering and clustering steps utilizing sequence-based MMseqs2 (Steinegger & Söding, 2017) and structure-based Foldseek (Van Kempen et al., 2024). It contains 588,318 structures with lengths ranging from 32 to 256 residues. For PROTEINAE training, we apply random global rotation to the protein structures as data augmentation. For structure reconstruction evaluation, we use the benchmark sets from CASP 14 and CASP 15. We use ATLAS Vander Meersche et al. (2024), a protein molecular dynamics (MD) dataset for downstream flexibility prediction. For our reconstruction and downstream results, we primarily use a default PROTEINAE configuration. The encoder and decoder DiT stacks have $L = 5$ layers and a token dimension $D = 256$. The bottleneck is configured with a length downsampling ratio $r = 1$ (i.e., no length downsampling) and compresses the dimension from $D = 256$ to a latent dimension $d = 8$. The PLDM is also based on a DiT architecture with 200M parameters, featuring $L = 15$ and $D = 768$. Different configurations and their impact on performance are explored in the ablation studies. The overall details can be found in Appendix A.4.

### 3.2 STRUCTURE RECONSTRUCTION

Table 1: Different protein autoencoders' structure reconstruction quality measured by RMSD ($\downarrow$). Lower is better. **Bold** indicates the best performance. *Note that ProToken and DPLM-2 can only process proteins under 2,048 residues.

| | Methods | CASP14 | | | CASP15 | |
|---|---|---|---|---|---|---|
| | | T | T-dom | oligo | TS-domains | oligo |
| $C_\alpha$ RMSD | CHEAP (Lu et al., 2024) | 11.15±9.88 | 8.99±9.18 | 9.93±10.55 | 10.22±11.23 | 9.22±12.57 |
| | ESM3 VQ-VAE (Hayes et al., 2025) | 1.02±1.82 | 0.66±0.42 | 3.08±7.39 | 1.23±1.26 | 1.94±2.43 |
| | ProToken* (Lin et al., 2023a) | 0.99±0.69 | 0.96±0.58 | 1.15±1.12 | 1.15±1.00 | 1.18±0.89 |
| | DPLM-2* (Wang et al., 2024) | 1.99±2.03 | 1.87±1.78 | 2.70±5.05 | 3.31±5.69 | 3.50±6.23 |
| | PROTEINAE | **0.23±0.15** | **0.22±0.11** | **0.31±0.22** | **0.28±0.20** | **0.37±0.50** |

A comprehensive benchmark analysis in Table 1 reveals that our proposed model exhibits clear and consistent superiority in the task of protein structure reconstruction. We give a comprehensive reviews of these baselines in Appendix B. PROTEINAE systematically outperforms all baseline methods, including prominent vector-quantized autoencoders, across the full suite of challenging targets from the CASP14 and CASP15 assessments. This robust outperformance holds true irrespective of the protein's structural class or complexity. The performance margin of PROTEINAE is particularly pronounced when reconstructing targets of high structural complexity. For challenging oligomeric assemblies, where many baseline models show a marked degradation in quality, our approach consistently maintains a high degree of fidelity. This capability underscores the robustness of our framework in capturing the intricate spatial arrangements and fine-grained geometric details that are often lost by competing methods. Collectively, these results strongly suggest that the diffusion autoencoder framework is more effective at modeling the manifold of protein structures than discrete, vector-quantized approaches. By circumventing the information bottlenecks inherent to tokenization, our model learns a richer and more expressive latent representation. This leads to significantly more accurate and reliable reconstructions, establishing a new state of the art in high-fidelity protein structure encoding. It is noting that CHEAP consistently records the poorest performance; this is an anticipated outcome, as it relies on encoding features from ESM2, thereby inheriting its limitations and being fundamentally capped by ESMFold's prediction accuracy.

### 3.3 DOWNSTREAM STRUCTURAL LATENT ANALYSIS

**Unconditional Backbone Generation** We assess the performance of PROTEINAE-PLDM on unconditional protein backbone generation against state-of-the-art Structure Diffusion Models (SDMs), Multi-Modal Language Models (MLLMs), and other Latent Diffusion Models (LDMs),

with results presented in Table 2. The details of PROTEINAE-PLDM methods are illustrated in Appendix A.3, including the flow training objectives and sampling within the PROTEINAE's latent space. The evaluation demonstrates that PROTEINAE-PLDM achieves state-of-the-art performance among latent-space generative models. Our approach outperforms those MLLM and LDM baselines, achieving high designability and diversity, where these prior latent-space methods have traditionally struggled. Furthermore, PROTEINAE-PLDM's performance extends beyond dominating its own category to rival that of classical SDMs, effectively closing the significant performance gap that has long separated latent- and structure-based diffusion models. PROTEINAE-PLDM can also provide a controllable trade-off with sampling temperature $\gamma$, where a higher temperature setting modestly decreases designability in exchange for a notable increase in structural diversity. Visual case studies (Figure 3D, purple indicates helix and yellow indicates sheet) further reveal that samples with high designability converge to common folds, while those with lower designability often represent novel yet physically plausible conformations. We also observe that the model's tendency to generate varied loop regions, reminiscent of AFDB structures (according to Huguet et al. (2024); Geffner et al. (2025), AFDB's overall designability is 33%, far from 100% designable), may contribute to these novel topologies and explain the small remaining designability gap to SDMs focused on canonical folds.

Table 2: Unconditional protein backbone generation performance among SDM, MLLM and LDM approaches. Baseline results are adopted from Yim et al. (2025). Metric details are illustrated in Appendix C.

| Type | Methods | Des (↑) | Div (↑) | DPT (↓) | Nov (↓) |
|---|---|---|---|---|---|
| SDM | RFdiffusion (Watson et al., 2023) | 96% | 247 | 0.43 | 0.71 |
| | ProteinSGM (Lee et al., 2023) | 49% | 122 | 0.37 | 0.51 |
| | FrameFlow PDB (Yim et al., 2023a) | 91% | 278 | 0.48 | 0.65 |
| | FrameFlow AFDB | 23% | 54 | 0.42 | 0.70 |
| MLLM | ESM3 (Hayes et al., 2025) | 61% | 127 | 0.37 | 0.84 |
| | DPLM-2 650M (Wang et al., 2024) | 63% | 130 | 0.37 | 0.72 |
| semi-LDM | LSD (Yim et al., 2025) | 69% | 203 | 0.46 | 0.74 |
| LDM | LatentDiff (Fu et al., 2024) | 17% | 34 | 0.51 | 0.73 |
| | PROTEINAE-PLDM $\gamma = 0.35$ | 93% | 204 | 0.36 | 0.70 |
| | PROTEINAE-PLDM $\gamma = 0.5$ | 86% | 228 | 0.35 | 0.66 |

**Generation Efficiency** We evaluate the generation efficiency of PROTEINAE-PLDM against prominent structure diffusion models, RFDiffusion (Watson et al., 2023) and multi-modal protein language model DPLM-2 650M Wang et al. (2024). The comparison is based on the *average* sampling time and GPU memory required to generate backbones of 200 residues with a batch size of 5 on a single 80G A100 GPU. As illustrated in Fig. 3C, PROTEINAE-PLDM demonstrates substantially higher efficiency than both baselines. It achieves the lowest sampling time (∼1.6 seconds) while consuming the least GPU memory (∼0.3 GB). RFDiffusion is the most computationally demanding, requiring ∼15 seconds and ∼5 GB of memory. DPLM-2 exhibits intermediate performance, with a sampling time of ∼3 seconds and memory usage of ∼1 GB. The remarkable efficiency of PROTEINAE-PLDM is primarily attributed to its dimension bottleneck design and the elimination of triangular attention. This allows the PLDM to operate entirely within a compact, low-dimensional latent space, bypassing the complex geometric or physical constraints inherent to direct structure generation and thereby drastically reducing the computational burden.

**Effectiveness Analysis** To evaluate the downstream effectiveness of the latent space learned by PROTEINAE, we benchmarked its performance on physicochemical property prediction. Following the setup from StructTokenBench (Yuan et al., 2025), we predicted residue-level structural flexibility (RMSF and B-factor) on the ATLAS dataset. The training details are illustrated in Appendix A.4. As shown in Table 3, PROTEINAE achieves the highest scores across all evaluated tasks and splits, demonstrating a clear advantage over the baseline models. In the task of predicting FlexRMSF, PROTEINAE attains the highest Spearman's $\rho$ correlation on both fold (close level) and superfamily (more remote) splits, improving upon ESM3 by over 10%. A similar trend is observed in the FlexBFactor prediction, where PROTEINAE again leads in performance across both splits. This

Table 3: Physicochemical Property Prediction (Spearman's $\rho\%$) on ATLAS. Baseline results are adopted from StructTokenBench (Yuan et al., 2025).

| Task | Split | Model | | | | | |
|------|-------|-------|-------|------|-----------|-----------|----------|
| | | FoldSeek | ProTokens | ESM3 | Vanilla VQ | AminoAseed | PROTEINAE |
| FlexRMSF | Fold | 15.35 | 13.81 | 44.53 | 44.22 | 44.63 | **45.36** |
| | SupFam | 11.99 | 7.62 | 39.68 | 39.08 | 40.99 | **44.71** |
| FlexBFactor | Fold | 4.17 | 6.67 | 23.60 | 22.32 | 21.30 | **30.87** |
| | SupFam | 6.97 | 5.47 | 25.80 | 23.73 | 21.76 | **26.54** |

indicates that the continuous representations learned by PROTEINAE have effectively captured the generalizable principles governing protein geometry and dynamics.

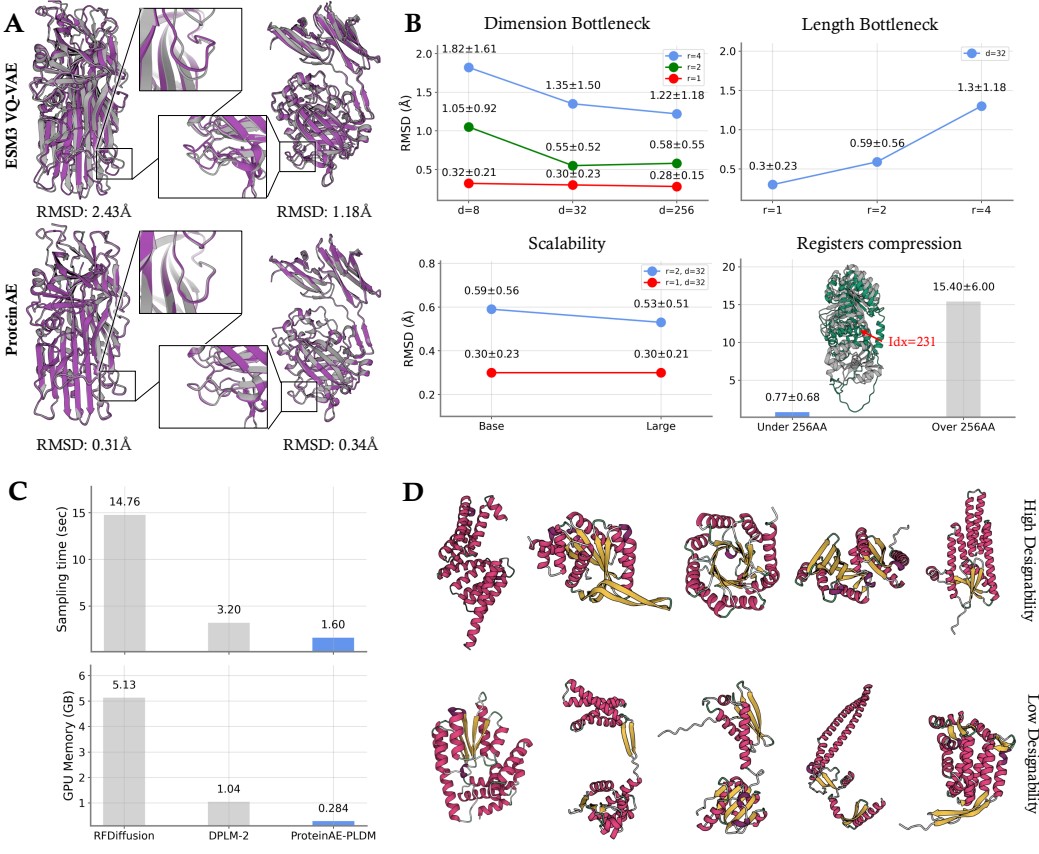

Figure 3: **Summary of PROTEINAE reconstruction, generation, and model architecture analysis.** (A) Visual comparison of protein structure reconstruction quality between PROTEINAE and ESM3 VQ-VAE. (B) Ablation studies on key architectural components: the impact of dimension and length bottlenecks, evaluation of model scalability with increased parameters, and analysis of using DiT registers for latent compression. (C) Generation efficiency comparison of PROTEINAE-PLDM, RFDiffusion, and DPLM-2, including average sampling time and GPU memory usage. (D) Visual examples of protein backbone structures generated unconditionally by PROTEINAE-PLDM.

## 3.4 ABLATION STUDY

In this section, we explore the influences of different designs of PROTEINAE. Specifically, we conduct experiments to investigate the impact of the protein length and dimension bottlenecks, evaluate

the model scalability, and analyze the effectiveness of using registers as an alternative strategy for latent representation. All the results are conducted on CASP15 TS-domains.

**Protein Length & Dimension Bottleneck**  We depict the reconstruction quality of protein length and dimension bottleneck in Fig. 3B. We choose 3 downsampling ratios $r = 1, 2, 4$, where $r = 1$ signifies no length downsampling. We also investigate 3 bottleneck dimensions $d = 8, 32, 256$, where $d = 256$ corresponds to no dimension reduction relative to the full feature dimension. As shown in Fig. 3B, both length and dimension compression impact reconstruction accuracy. Increasing the dimension bottleneck (decreasing $d$) generally leads to a moderate increase in RMSD, as less information is preserved in the latent space. However, increasing the length downsampling ratio (increasing $r$) results in a significantly larger degradation in reconstruction quality, with RMSD increasing substantially as $r$ goes from 1 to 4. This indicates that preserving the sequential length dimension is more critical for accurate protein backbone reconstruction than maintaining a large feature dimension.

**Scalability**  We study the scalability of PROTEINAE on two variants: PROTEINAE-Base ($\sim$20M parameters) and PROTEINAE-Large ($\sim$100M parameters). The results, showing reconstruction quality (RMSD) for these variants under different bottleneck configurations ($r = 1, d = 32$ and $r = 2, d = 32$), are presented in the "Scalability" plot in Fig. 3B. As the model size increases from Base to Large, we observe a slight improvement in reconstruction quality (decrease in RMSD) for the configuration with length downsampling ($r = 2, d = 32$). For the configuration without length downsampling ($r = 1, d = 32$), the RMSD remains low and stable across both model sizes. This indicates that PROTEINAE exhibits positive scalability, where increasing model capacity leads to comparable or slightly improved performance, particularly noticeable when the task is more challenging due to bottlenecking (e.g., with $r = 2$). The plot also reinforces that maintaining the original protein length ($r = 1$) is beneficial for reconstruction accuracy regardless of the model size.

**Conditional Decoding using Registers**  Beyond the bottleneck compression strategy for obtaining the latent representation $z$, we also explore using the learnable register tokens to compress the structures to fixed collection. The registers are concatenated to the input sequence representation and participate in attention computations (as mentioned in Section 2). Recent works, such as FlexTok (Bachmann et al., 2025), have demonstrated that these tokens can effectively serve as compact latent representations for images. Inspired by this, we investigate a variant, termed PROTEINAE-Register, where the registers from the encoder ($z^{\text{reg}}$) are directly used as the registers in the decoder, enabling conditional generation based on these tokens instead of the bottlenecked latent $z$. We evaluate the reconstruction quality of PROTEINAE-Register, with results depicted in Fig. 3B. The plot shows that PROTEINAE-Register achieves good reconstruction quality for protein lengths up to 256 residues, which corresponds to the maximum length in our training dataset. However, performance degrades dramatically for structures exceeding this training length limit. Fig. 3B also visualizes an example of this failure mode, showing the reconstruction breaking down abruptly around residue 231 for a longer protein. These results highlight a limitation of directly applying a register-based compression strategy to variable-length protein sequences. The success of registers as latent representations in vision tasks may be partly attributable to the fixed size of image tokens after patchification, a characteristic not present in native protein sequences of arbitrary length. This suggests that for protein structures, alternative compression mechanisms like length and dimension bottlenecks are more robust to variable input sizes, particularly for handling lengths beyond the training distribution. We will also explore a better way to encode protein structures into fixed-length vectors in the future.

## 4  CONCLUSION AND LIMITATIONS

We introduced PROTEINAE, a novel protein diffusion autoencoder that maps protein backbone structures into a continuous, compact latent space using a non-equivariant Diffusion Transformer architecture and a simple flow matching objective. It avoids the operations on intricate $SE(3)$ and discrete representations inherent in existing methods. Building on this learned latent space, we further developed PLDM for structure generation. Our experiments demonstrate that PROTEINAE achieves high reconstruction quality and enables efficient protein structure generation. Despite its capabilities, PROTEINAE still has several limitations that warrant future investigation. For instance, PROTEINAE is currently limited to modeling protein monomers and cannot model other biomolecules

such as ligands, DNA, or RNA. Besides, the generative performance of PLDM is not yet significantly outperforming state-of-the-art structure-based generative models. Finally, PLDM exhibits challenges related to sequence length handling, which can lead to structural collapse or unrealistic geometries for certain residues in the generated outputs. We plan to address these limitations and explore extensions to PROTEINAE in future work.

## 5 REPRODUCIBILITY STATEMENT

We demonstrate the dataset and model configuration in Sec. 3.1, the model architecture details in Sec. 2 and Appendix A.1, the training details of different tasks in Appendix A.4.

## 6 THE USE OF LARGE LANGUAGE MODELS (LLMs)

In the course of writing this paper, a large language model (LLM) was used as a tool to refine the language and grammar of the background section. We meticulously reviewed and revised all content to take full responsibility for the final text. Additionally, we consulted an LLM for the implementation of certain metrics. The underlying code for these metrics was carefully examined and validated to ensure its correctness and appropriateness for our study.

### ACKNOWLEDGMENTS

The work described in this paper was supported in part by the Research Grants Council of the Hong Kong Special Administrative Region, China, under Project T45-401/22-N. We thank Prof. Yaosen Min for valuable discussion.

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

# A  ARCHITECTURE DETAILS

## A.1  DIFFUSION TRANSFORMERS AND ALL-ATOM ATTENTION

The yellow background indicates the differences between PROTEINAE and AlphaFold3.

---

**Algorithm 1** Attention with pair bias and mask

---

1: **function** ATTENTIONPAIRBIAS($\{\mathbf{s}_i\}, \{\mathbf{c}_i\}, \{\mathbf{p}_{ij}\}, \{\beta_{ij}\}, N_{\text{head}}$)
2:   $\mathbf{a}_i \leftarrow \text{AdaLN}(\mathbf{s}_i, \mathbf{c}_i)$
3:   $\mathbf{q}_i^h = \text{Linear}(\mathbf{s}_i)$
4:   $\mathbf{k}_i^h, \mathbf{v}_i^h = \text{LinearNoBias}(\mathbf{s}_i)$
5:   $\mathbf{k}_i^h, \mathbf{v}_i^h = \text{LayNorm}(\mathbf{k}_i^h), \text{LayNorm}(\mathbf{v}_i^h)$
6:   $\boxed{\mathbf{k}_i^h, \mathbf{v}_i^h = \text{RoPE}(\mathbf{k}_i^h, \mathbf{v}_i^h)}$
7:   $\mathbf{b}_{ij}^h \leftarrow \text{Linear}(\mathbf{p}_{ij}) + \beta_{ij}$
8:   $\mathbf{g}_i^h \leftarrow \text{sigmoid}(\text{Linear}(\mathbf{a}_i))$
                          ▷ Attention
9:   $A_{ij}^h \leftarrow \text{softmax}_j \left( \frac{1}{\sqrt{c}} (\mathbf{q}_i^h)^T \mathbf{k}_j^h + \mathbf{b}_{ij}^h \right)$
10:   $\mathbf{s}_i \leftarrow \text{LinearNoBias} \left( \text{concat}_h \left( \mathbf{g}_i^h \circ \sum_j A_{ij}^h \mathbf{v}_j^h \right) \right)$
11:   **return** $\{\mathbf{s}_i\}$
12: **end function**

---

**Algorithm 2** Diffusion Transformer

---

1: **function** DIFFUSIONTRANSFORMER($\{\mathbf{s}_i\}, \{\mathbf{c}_i\}, \{\mathbf{p}_{ij}\}, \{\beta_{ij}\}, N_{\text{block}}, N_{\text{head}}$)
2:   **for all** $n \in \{1, \ldots, N_{\text{block}}\}$ **do**
3:     $\{\mathbf{s}_i\} = \text{AttentionPairBias}(\{\mathbf{s}_i\}, \{\mathbf{c}_i\}, \{\mathbf{p}_{ij}\}, \{\beta_{ij}\}, N_{\text{head}})$
4:     $\mathbf{a}_i \leftarrow \mathbf{s}_i + \text{ConditionedTransitionBlock}(\mathbf{s}_i, \mathbf{c}_i)$
5:   **end for**
6:   **return** $\{\mathbf{s}_i\}$
7: **end function**

---

**Algorithm 3** Atom Transformer

---

1: **function** ATOMTRANSFORMER($\{\mathbf{q}_l\}, \{\mathbf{c}_l\}, \{\mathbf{p}_{lm}\}, N_{\text{block}} = 3, N_{\text{head}}, N_{\text{queries}} = 32, N_{\text{keys}} = 128, \mathcal{S}_{\text{subset centres}} = \{15.5, 47.5, 79.5, \ldots\}$)
2:   $\beta_{lm} = \begin{cases} 0 & \text{if } ||l - c|| < N_{\text{queries}}/2 \wedge ||m - c|| < N_{\text{keys}}/2 \quad \forall c \in \mathcal{S}_{\text{subset centres}} \\ -10^{10} & \text{else} \end{cases}$
3:   $\{\mathbf{q}_l\} \leftarrow \text{DiffusionTransformer}(\{\mathbf{q}_l\}, \{\mathbf{c}_l\}, \{\mathbf{p}_{lm}\}, \{\beta_{lm}\}, N_{\text{block}}, N_{\text{head}})$
4:   **return** $\{\mathbf{q}_l\}$
5: **end function**

---

---

**Algorithm 4** All-Atom Attention Encoder

---

1: **function** ALLATOMATTNENCODER($\{\mathbf{f}_i^s\}, \{\mathbf{x}\}, \{\mathbf{s}\}, \{\mathbf{c}\}, \{\mathbf{p}_{ij}\}, c_{\text{atom}} = 64, c_{\text{atompair}} = 16,$
   $c_{\text{token}} = 256$ )
2: $\quad \mathbf{c}_i \leftarrow \text{LinearNoBias}(\text{concat}(\mathbf{f}_i^{\text{ref\_pos}}, \mathbf{f}_i^{\text{ref\_charge}}, \mathbf{f}_i^{\text{ref\_mask}}, \mathbf{f}_i^{\text{ref\_element}}, \mathbf{f}_i^{\text{ref\_atom\_name\_chars}}))$
3: $\quad \mathbf{d}_{lm} \leftarrow \mathbf{f}_l^{\text{ref\_pos}} - \mathbf{f}_m^{\text{ref\_pos}}$
4: $\quad \mathbf{v}_{lm} \leftarrow \mathbf{f}_{lm}^{\text{ref\_space\_uid}} \circ \mathbf{d}_{lm}$
5: $\quad \mathbf{p}_{lm} \mathrel{+}= \text{LinearNoBias}(\mathbf{d}_{lm}) \cdot \mathbf{v}_{lm}$
6: $\quad \mathbf{p}_{lm} \mathrel{+}= \text{LinearNoBias}\left(\frac{1}{1+||\mathbf{d}_{lm}||^2}\right) \cdot \mathbf{v}_{lm}$
7: $\quad \mathbf{p}_{lm} \mathrel{+}= \text{LinearNoBias}(\mathbf{v}_{lm}) \cdot \mathbf{v}_{lm}$
8: $\quad \mathbf{q}_l \leftarrow \mathbf{c}_l$
9: $\quad \boxed{\mathbf{c}_l \mathrel{+}= \text{LinearNoBias}(\text{LayerNorm}(\mathbf{c}_{\text{tok\_idx}(l)}))}$
10: $\quad \mathbf{p}_{lm} \mathrel{+}= \text{LinearNoBias}(\text{LayerNorm}(\mathbf{p}_{\text{tok\_idx}(l),\text{tok\_idx}(m)}))$
11: $\quad \mathbf{q}_l \mathrel{+}= \text{LinearNoBias}(\mathbf{r}_l)$
12: $\quad \mathbf{p}_{lm} \mathrel{+}= \text{LinearNoBias}(\text{relu}(\mathbf{c}_l)) + \text{LinearNoBias}(\text{relu}(\mathbf{c}_m))$
13: $\quad \mathbf{p}_{lm} \mathrel{+}= \text{LinearNoBias}(\text{relu}(\text{LinearNoBias}(\text{relu}(\text{LinearNoBias}(\text{relu}(\mathbf{p}_{lm}))))))$
14: $\quad \{\mathbf{q}_l\} \leftarrow \text{AtomTransformer}(\{\mathbf{q}_l\}, \{\mathbf{c}_l\}, \{\mathbf{p}_{lm}\}, N_{\text{block}} = 3, N_{\text{head}} = 4)$
15: $\quad \mathbf{s}_l \leftarrow \text{mean}_{i \in \{1 \dots N_{\text{atoms}}\}, i \rightarrow \text{tok\_idx}(l)}(\text{relu}(\text{LinearNoBias}(\mathbf{q}_i)))$
16: $\quad \mathbf{q}_l^{\text{skip}}, \mathbf{c}_l^{\text{skip}}, \mathbf{p}_{lm}^{\text{skip}} \leftarrow \mathbf{q}_l, \mathbf{c}_l, \mathbf{p}_{lm}$
17: $\quad$ **return** $\{\mathbf{s}_l\}, \{\mathbf{q}_l^{\text{skip}}\}, \{\mathbf{c}_l^{\text{skip}}\}, \{\mathbf{p}_{lm}^{\text{skip}}\}$
18: **end function**

---

**Algorithm 5** All-Atom Attention Decoder

---

1: **function** ALLATOMATTNDECODER($\{\mathbf{s}_i\}, \{\mathbf{q}_l^{\text{skip}}\}, \{\mathbf{c}_l^{\text{skip}}\}, \{\mathbf{p}_{lm}^{\text{skip}}\}$)
2: $\quad \mathbf{q}_l \leftarrow \text{LinearNoBias}(\mathbf{s}_{\text{tok\_idx}(l)}) + \mathbf{q}_l^{\text{skip}}$
3: $\quad \{\mathbf{q}_l\} \leftarrow \text{AtomTransformer}(\{\mathbf{q}_l\}, \{\mathbf{c}_l^{\text{skip}}\}, \{\mathbf{p}_{lm}^{\text{skip}}\}, N_{\text{block}} = 3, N_{\text{head}} = 4)$
4: $\quad v \leftarrow \text{LinearNoBias}(\text{LayerNorm}(\mathbf{q}_l))$
5: $\quad$ **return** $\{v\}$
6: **end function**

---

## A.2 PROTEINAE FLOW DECODING

PROTEINAE decodes reconstructed protein structures by simulating the learned ODE velocity field, conditioned on the latent representation $z$:

$$\frac{dx_t}{dt} = v_t^\theta(x_t, t, z) \tag{10}$$

Starting from a noise sample $x_T$, the ODE is integrated from $t = 1$ to $t = 0$ to obtain the denoised structure $x_0$. In practice, we integrate the ODE using a numerical solver (e.g., Euler method) with a small number of discretization steps to reduce computational cost. The latent representation $z$ acts as a powerful condition during decoding, guiding the structure generation process. It serves a role analogous to sequence features (like MSAs) in models such as AlphaFold3, enabling more deterministic structure generation conditioned on the learned latent.

### A.3 PROTEIN LATENT DIFFUSION MODELING (PLDM)

Protein Latent Diffusion Modeling (PLDM) is our framework for generating novel latent representations. We employ flow matching as the generative approach in the latent space, which learns a continuous transformation to map samples from a simple distribution (e.g., Gaussian) to the target latent distribution. Unlike structure-based generative models that operate on the SE(3) manifold of protein structures, our PLDM operates purely on the compact latent representation $z$ learned by the PROTEINAE, as illustrated in Fig. 2c.

**PLDM Training Objective**  PLDM is trained using a flow-matching loss applied to the latent representation $z \in \mathbb{R}^{N_{\text{down}} \times d}$. The model $v^\phi$ is trained to predict the velocity field in the latent space. Given a clean structure $x_1$ from the data distribution $p_{\text{ds}}(x)$, its latent representation is $\mathcal{E}(x_1)$. We define a linear path in the latent space between a noise sample $z_0 \sim \mathcal{N}(0, I)$ and $\mathcal{E}(x_1)$: $z_t = (1-t)z_0 + t\mathcal{E}(x_1)$. The target velocity for this path is $\frac{dz_t}{dt} = \mathcal{E}(x_1) - z_0$. The PLDM is parameterized by $\phi$, and its training objective is:

$$\min_\phi \mathbb{E}_{x_1 \sim p_{\text{ds}}(x), z_0 \sim \mathcal{N}(0,I), t \sim p(t)} \left[ \frac{1}{N_{\text{down}}} \left\| v^\phi(z_t, t) - (\mathcal{E}(x_1) - z_0) \right\|_2^2 \right], \tag{11}$$

where $v^\phi(z_t, t)$ is the velocity predicted by the PLDM network given the noisy latent sample $z_t$ and timestep $t$. The expectation is taken over data samples $x_1$, noise samples $z_0$, and timesteps $t$ sampled from same $p(t)$.

**PLDM Sampling**  Following observations from Proteina (Geffner et al., 2025) regarding the limitations of directly sampling the full learned distribution, we perform sampling from the latent space using an SDE-based schedule. The SDE for generating latent samples $\hat{z}$ is defined as:

$$dz_t = v^\phi(z_t, t) \cdot dt + g(t)s^\phi(z_t, t)dt + \sqrt{2g(t)\gamma}\, d\mathcal{W}_t, \tag{12}$$

where $v^\phi(z_t, t)$ is the velocity predicted by the trained PLDM model, $g(t)$ is the diffusion coefficient, $s^\phi(z_t, t)$ is the score function derived from the predicted velocity (specifically, $s^\phi(z_t, t) \approx \nabla_{z_t} \log p_t(z_t)$), $\gamma$ is a noise scale parameter, and $d\mathcal{W}_t$ is a standard Wiener process. Starting from a sample from the Gaussian distribution $z_T \sim \mathcal{N}(0, I)$, we integrate this SDE numerically from $t = 1$ to $t = 0$ to obtain the generated latent $\hat{z}$. Finally, this generated latent $\hat{z}$ is decoded into a protein structure $\hat{x}$ using the trained PROTEINAE flow decoder: $\hat{x} = \mathcal{D}(\hat{z})$ (Eq. 10).

## A.4 TRAINING DETAILS

Table 4: Training and model hyperparameters for PROTEINAE, PLDM, and Flexibility Prediction.

| Parameter | PROTEINAE | PLDM | Flexibility Prediction |
|---|---|---|---|
| *Training Details* | | | |
| GPUs | $4 \times 80G$ A100 | $8 \times 80G$ A100 | $1 \times 80G$ A100 |
| Batch size per GPU | 8 | 64 | 8 |
| Global batch size | 32 | 512 | 8 |
| Training duration | 10 epochs | 200,000 steps | 300 epochs |
| Optimizer | Adam | Adam | Adam |
| Learning rate | $1 \times 10^{-4}$ | $1 \times 10^{-4}$ | $1 \times 10^{-4}$ |
| Loss function | Flow-matching | | MSE |
| *Model Architecture* | | | |
| Token dimension | 256 | 768 | – |
| Layers | 5 | 15 | – |
| Attention heads | 8 | 12 | – |
| Condition dimension | 256 | 512 | – |
| Timestep embedding size | 256 | – | – |
| Sequence index embedding | 128 | – | – |
| Downsampling factor ($r$) | 1 | – | – |
| Dimension bottleneck | 8 | – | – |
| *All-Atom Attention Module (PROTEINAE only)* | | | |
| Atom dimension | 64 | – | – |
| Atom pair dimension | 6 | – | – |
| Attention mask (query/key) | 32 / 128 | – | – |
| *Flexibility Prediction Model* | | | |
| Architecture | MLP | | |
| Dimension | 512 | | |
| Activation function | Relu | | |

# B  BACKGROUND

**Diffusion models**  The foundational diffusion model concepts introduced by (Sohl-Dickstein et al., 2015) and significantly advanced with denoising diffusion probabilistic models (DDPMs) (Ho et al., 2020)—which iteratively add noise to data and then learn to reverse the process. More recently, flow matching has emerged as a powerful and efficient alternative (Lipman et al., 2022; Liu et al., 2022; Albergo et al., 2023). These flow-based methods directly learn a vector field, often an Ordinary Differential Equation (ODE), to transform a simple noise distribution into a complex data distribution, frequently enabling simulation-free training and faster sampling. Both diffusion and flow matching paradigms can be further extends to general manifolds (Chen & Lipman, 2023) as well as discrete data (Cheng et al., 2024; Austin et al., 2021). They have found extensive applications in de novo protein design, with models like RFDiffusion, FrameDiff (initially diffusion-based) and their successor RFdiffusion2, FrameFlow (which incorporates flow matching for training). Some work (Campbell et al., 2024; Li et al., 2024) has also attempted the co-design of sequences and structures. More recently, Proteina (Geffner et al., 2025) attempted to model coordinates directly using a non-equivariant architecture on E(3), but they only work with $C_\alpha$.

**Protein Autoencoders**  Autoencoders play a significant role in learning compact and meaningful representations for both protein sequences and structures, which are vital for efficient generative modeling and other downstream tasks. While protein sequence autoencoders are widely used, often leveraging features like MSAs or ESM embeddings (Detlefsen et al., 2022; Lu et al., 2024; 2025; Chen et al., 2024), the development of autoencoders for protein *structures* is a more recent area of research. A notable example that bridges these areas is CHEAP (Lu et al., 2024). This model operates on protein language model features from ESM2, which are then used to predict structure via ESMFold's structure module. While this method can yield valuable structural information, its encoding capability is inherently dependent on, and thus limited by, ESMFold's prediction accuracy.

In parallel, pioneering efforts have focused on autoencoding the protein structure directly. These models tackle the challenge of converting continuous 3D atomic coordinates into discrete tokens. Key examples include the ProToken Lin et al. (2023a), ESM3 VQ-VAE tokenizer (Hayes et al., 2025) and the DPLM-2 lookup-free quantization (LFQ) tokenizer (Wang et al., 2024; 2025). By enabling this conversion, they allow for the joint modeling of protein sequences and structures within large generative frameworks. Subsequent work, such as AminoAseed (Yuan et al., 2025), has further explored improvements in the codebook design for these discrete structure representations. However, a common challenge for these tokenization-based autoencoders is the inherent information loss that occurs during the discretization process. This makes it difficult to fully capture the complexity and subtle geometric details present in the continuous 3D space of a protein structure.

**Latent Space Generative Models for Proteins**  While diffusion models operating directly on atomic coordinates or torsion angles have achieved remarkable success, they often suffer from high computational costs due to the vast exploration space of 3D protein structures. To address this, recent works have investigated diffusing in a compressed latent space. For instance, Fu et al. (2024) proposed LatentDiff, which employs an SE(3)-equivariant graph autoencoder to map protein backbones into a condensed latent representation. By performing the diffusion process within this lower-dimensional latent space rather than the original coordinate space, LatentDiff significantly reduces computational complexity, achieving sampling speeds orders of magnitude faster than frame-based approaches like FrameDiff Yim et al. (2023b) or RFdiffusion Watson et al. (2023).

Table 5: Comparison of different protein autoencoder architectures, highlighting their input modalities, representation types, and operational scopes.

| Methods | Input Modality | Representation Type | Operational Scope | Compression Strategy |
|---|---|---|---|---|
| **ProteinAE** | Atom coordinates (Structure) | Continuous | All proteins | ✓ Direct structural compression |
| **CHEAP** | ESM2 features (Sequence) | Continuous | Proteins ESMFold predicts well | ✓ Sequence feature compression |
| **ProToken, ESM3 VQ-VAE & DPLM-2 LFQ** | Backbone Frame (Structure) | Discrete | All proteins (with 2,048 length constraints for ProToken and DPLM-2 LFQ) | ✗ |

## B.1 BACKBONE-LEVEL RECONSTRUCTION QUALITY

Table 6: Different protein autoencoders' structure reconstruction quality measured by RMSD (↓). Lower is better. **Bold** indicates the best performance. *Proteins missing backbone atoms are ignored.* *Note that ProToken and DPLM-2 can only process proteins under 2,048 residues.

| | Methods | CASP14 | | | CASP15 | |
|---|---|---|---|---|---|---|
| | | T | T-dom | oligo | TS-domains | oligo |
| | CHEAP | 11.16±10.09 | 4.71±5.21 | 11.10±11.95 | 10.98±12.44 | 8.24±14.12 |
| Backbone RMSD | ESM3 VQ-VAE | 1.28±2.32 | 0.66±0.42 | 3.11±7.41 | 1.25±1.28 | 2.47±2.27 |
| | ProToken* | 1.14±0.66 | 1.09±0.16 | 1.55±1.46 | 1.29±0.67 | 1.33±0.70 |
| | DPLM-2* | 1.94±1.99 | 1.47±0.42 | 3.81±7.20 | 4.58±6.62 | 3.83±6.90 |
| | PROTEINAE | **0.51±1.56** | **0.23±0.11** | **0.31±0.27** | **0.31±0.21** | **0.43±0.51** |

## C  METRIC DETAILS AND ADDITIONAL EXPERIMENTS

We complete the details of protein unconditional generation in Sec. 3.

**Designability**  Protein designability is assessed based on whether a protein backbone structure can be generated from a specific amino acid sequence that folds into that structure. Following the method of FrameDiff (Yim et al., 2023a), eight sequences are generated per backbone using ProteinMPNN (Dauparas et al., 2022) with a sampling temperature of 0.1. Structures are predicted using ESMFold (Lin et al., 2023b). The Root Mean Square Deviation (RMSD) between the predicted and original structures is calculated. A sample is deemed designable if its lowest RMSD, termed self-consistency RMSD (scRMSD), is $\leq 2$Å. The overall designability score is the fraction of designable samples.

**Designable Pairwise TM-score (DPT)**  This is the first of two methods for evaluating diversity, based on the methodology by (Bose et al., 2023). It involves calculating pairwise TM-scores (Zhang & Skolnick, 2004) among all designable samples for each specified protein length, and then aggregating the averages across different lengths. TM-scores range from 0 to 1, where higher scores indicate greater structural similarity. Therefore, lower TM-scores are preferred for this metric as they suggest greater diversity.

**Diversity (Cluster)**  The second diversity measure follows the approach by FrameDiff (Yim et al., 2023b). Designable backbones are clustered using Foldseek (Van Kempen et al., 2024) based on a TM-score threshold of 0.5. Diversity is calculated as the ratio of the total number of clusters to the number of designable samples:

$$\text{Diversity (Cluster)} = \frac{\text{Number of designable clusters}}{\text{Number of designable samples}} \tag{13}$$

In this case, higher scores are preferable, indicating that the designable samples form a larger number of distinct clusters, thus representing greater diversity.

**Novelty**  Novelty quantifies how distinct a model's generated structures are compared to structures present in predefined reference databases (AFDB and PDB in our work). For each designable structure generated by the model, its TM-score is computed against every structure in the reference set, and the maximum TM-score is recorded. The novelty score is the average of these maximum TM-scores over all designable samples. Lower scores are considered better, as they suggest the generated structures are less similar to known structures in the reference sets, implying higher novelty.

**Evaluation Details**  For protein unconditional generation, we sample 10 proteins of each length between 60-128, resulting in 690 samples in total. It is noting that the designability is not a accurate indicator of how well a generative model matches the training distribution since the training dataset is far from 100% designable (Huguet et al., 2024).

## D IMPACT OF LENGTH BOTTLENECK ON GENERATION

We further investigate the impact of imposing a length bottleneck to PROTEINAE on the PLDM generation process.

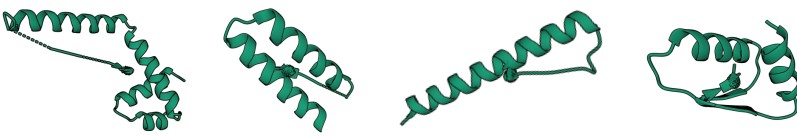

Figure 4: Illustration of structural collapse observed during protein generation by PLDM under length bottleneck conditions.

As depicted in Fig. 4, we observe that specific regions of the generated protein structures collapse towards the center, indicating a failure to maintain realistic local or global geometry under these conditions. We hypothesize that this phenomenon is attributed to the use of nearst interpolation during the upsampling process within PROTEINAE or the padding value in the downsampling training. We plan to investigate this issue in detail in future work.

## E   PROTEINAE V2 PREVIEW

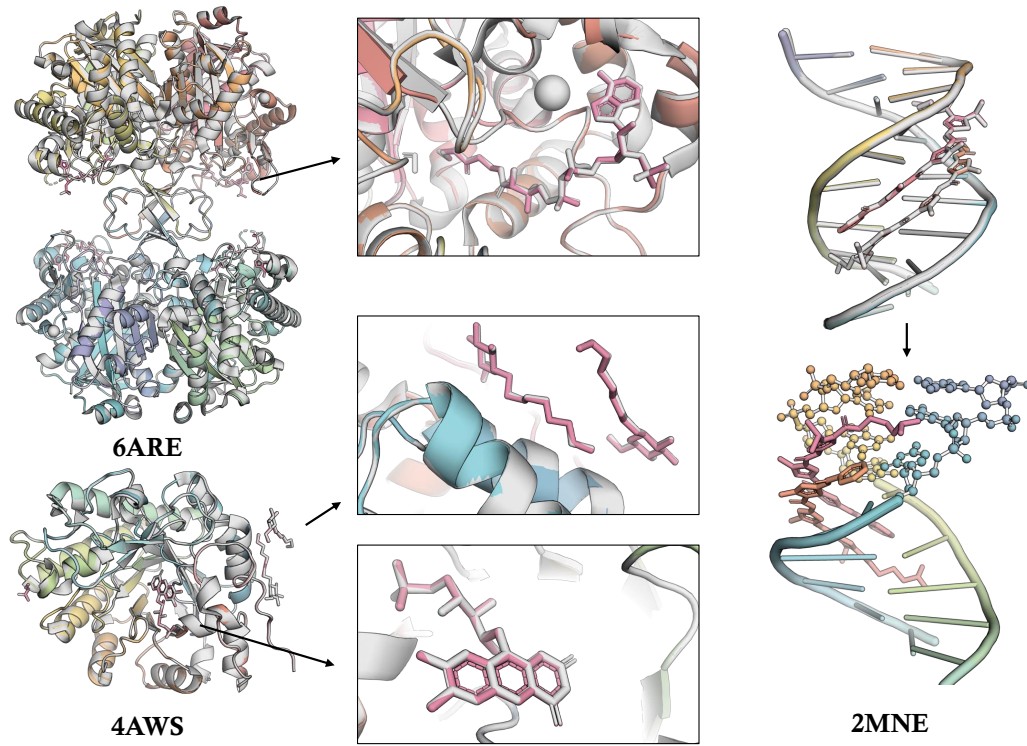

**6ARE**

**4AWS**

**2MNE**

Figure 5: **PROTEINAE v2 achieves high-fidelity all-atom reconstruction of multi-modal complexes.** The reconstructed structures are depicted in color, while the ground truth complexes are shown in grey for comparison.

