# OpenReview forum: "ProteinAE: Protein Diffusion Autoencoders for Structure Encoding"
_ICLR.cc/2026/Conference — ICLR 2026 Poster_

### Official Review · Reviewer_qmAJ · 2025-10-26

**Soundness:** 3
**Presentation:** 3
**Contribution:** 3
**Rating:** 6
**Confidence:** 3

**Summary:**

This paper proposed PROTEINAE, a protein diffusion autoencoder that learns compact, continuous representations of protein structures directly from 3D backbone coordinates in Euclidean space (E(3)).

Unlike prior approaches that depend on SE(3)-equivariant models, discrete tokenization, or multiple loss functions, PROTEINAE simplifies training with a single flow-matching objective and a non-equivariant Diffusion Transformer (DiT) architecture. The model efficiently encodes protein backbones into a latent space through a bottleneck design, enabling high-fidelity reconstruction and scalable downstream modeling. On benchmarks such as CASP14 and CASP15, PROTEINAE achieves state-of-the-art reconstruction accuracy, substantially outperforming discrete and equivariant autoencoders.

The authors argue that strict equivariance may not be essential for effective protein representation learning, citing recent advances such as AlphaFold3 and Proteina, which also employ non-equivariant architectures. However, the paper could benefit from a more comprehensive discussion of related work on 3D protein autoencoder design, as well as a deeper theoretical justification for the choice of a non-equivariant model.

**Strengths:**

## Strengths

- This paper introduces a non-equivariant diffusion autoencoder that learns continuous protein structure representations directly from 3D coordinates, simplifying model design and training compared to SE(3)-equivariant or discrete approaches. But as mentioned before, the paper could benefit from a deeper theoretical justification for the choice of a non-equivariant model.

- The proposed method achieves state-of-the-art reconstruction accuracy on CASP14/15 and competitive structure generation quality relative to more complex diffusion models.

- Computational efficiency: The latent diffusion framework offers good speed and memory advantages over previous diffusion methods.

**Weaknesses:**

## Weaknesses:

- This paper lacks sufficient discussion of prior work on 3D protein backbone structure autoencoder design. For example, the model LatentDiff is compared in Table 2, but it is not discussed in the related-work section. It shares similar design ideas with the proposed method, including an autoencoder that reduces the latent diffusion modelling burden, shrinks the modelling space in terms of length, and offers computational efficiency.

- The authors argue that strict equivariance may not be essential for effective protein representation learning by citing recent advances such as AlphaFold3 and Proteina, which also employ non-equivariant architectures. However, the paper would benefit from a deeper theoretical justification for choosing a non-equivariant model.

**Questions:**

As listed above in weaknesses.

---

> ### Author Response · Authors · 2025-11-19
> **Official Comment by Authors**
>
> ## Response to Reviewer qmAJ
>
> > Comment 1: This paper lacks sufficient discussion of prior work... For example, the model LatentDiff is compared in Table 2, but it is not discussed in the related-work section.
> >
>
> **Response: We apologize for this oversight and thank the reviewer for bringing LatentDiff to our attention.** We have added the following discussion regarding LatentDiff to the "Related Work" section:
>
> **Latent Space Generative Models for Proteins.**
> While diffusion models operating directly on atomic coordinates or torsion angles have achieved remarkable success, they often suffer from high computational costs due to the vast exploration space of 3D protein structures. To address this, recent works have investigated diffusing in a compressed latent space. For instance, **Fu et al. (2023)** [1] proposed **LatentDiff**, which employs an SE(3)-equivariant graph autoencoder to map protein backbones into a condensed latent representation. By performing the diffusion process within this lower-dimensional latent space rather than the original coordinate space, LatentDiff significantly reduces computational complexity, achieving sampling speeds orders of magnitude faster than frame-based approaches like FrameDiff [2] or RFdiffusion [3].
>
> [1] Fu, Cong, et al. "A latent diffusion model for protein structure generation." Learning on graphs conference. PMLR, 2024.
>
> [2] Yim, Jason, et al. "SE (3) diffusion model with application to protein backbone generation." arXiv preprint arXiv:2302.02277 (2023).
>
> [3] Watson, Joseph L., et al. "De novo design of protein structure and function with RFdiffusion." Nature 620.7976 (2023): 1089-1100.

---

> ### Author Response · Authors · 2025-11-19
>
> > Comment 2: The authors argue that strict equivariance may not be essential... However, the paper would benefit from a deeper theoretical justification for choosing a non-equivariant model.
> >
>
> **Response: We appreciate the opportunity to clarify our motivations.** We agree that SE(3) representations have historically been significant in the protein domain. However, recent state-of-the-art works—such as **AlphaFold 3** [1] for structure prediction, **Proteina** [2] for design, **SCN** [3] for small molecule representation, and **RADM** [4] and **ADiT** [8] for generation—have increasingly adopted **scalable non-equivariant architectures**. This represents a shifting trend: non-equivariant models possess simpler structures, avoiding complex components like spherical harmonics used in SE(3) Transformers [5], which allows them to scale up parameters and training data more efficiently.
>
> Our perspective on the necessity of SE(3) equivariance distinguishes between two categories of tasks:
>
> 1. **Generation, Reconstruction, and Cross-modal Tasks (e.g., Folding/Inverse Folding):**
> For these tasks, strict equivariance is not mandatory. For example, AlphaFold inputs a sequence and outputs a 3D structure; the absolute orientation of the prediction is irrelevant (notably, the powerful AlphaFold 2 [6] utilizes an Invariant Point Attention (IPA) module).
> In generative modeling (e.g., Flow Matching), the goal is to map a simple distribution (like Gaussian) to a target distribution. Even if the target structure is rotated, the model can learn to map the noise to the distribution via optimal transport and data augmentation (random rotations/translations during training), a standard practice in image and point cloud domains. Similarly, for autoencoders, the priority is high-quality reconstruction given an input. While intrinsic equivariance is a "nice-to-have," we demonstrate experimentally that **ProteinAE learns approximate SE(3) equivariance** through data augmentation, without enforcing it architecturally.
> 2. **Physical Understanding and Force Fields (MLFF):**
> For physical tasks like Machine Learning Force Fields (MLFF), SE(3) equivariance is critical. MLFFs predict scalar energy and vector forces to drive molecular dynamics simulations. If a system rotates, the predicted force vectors *must* rotate exactly in kind to ensure physical validity. This ***cannot*** be easily solved by simple data augmentation.
> *Evidence:* Kreiman et al. [7] attempted to use non-equivariant Transformers for MLFF. As shown in their Table 1, a **1B parameter Transformer** significantly underperformed a **6M parameter EGNN** in force prediction, supporting our view that strict equivariance is essential for physics simulations but not "that" necessarily for structural generation/reconstruction.
>
> | Model | Energy MAE (meV) | Forces MAE (meV/Å) |
> | --- | --- | --- |
> | eSEN-sm-d 6M | 129.77 | 13.01 |
> | Transformer 1B | 117.99 | 18.35 |
>
> **References:**
>
> [1] Abramson, Josh, et al. "Accurate structure prediction of biomolecular interactions with AlphaFold 3." *Nature* 630.8016 (2024): 493-500.
>
> [2] Geffner, Tomas, et al. "Proteina: Scaling flow-based protein structure generative models." ICLR 2025.
>
> [3] Zitnick, Larry, et al. "Spherical channels for modeling atomic interactions." NeurIPS 2022: 8054-8067.
>
> [4] Ding, Yuhui, and Thomas Hofmann. "Scalable Non-Equivariant 3D Molecule Generation via Rotational Alignment." ICML 2025.
>
> [5] Baek, Minkyung, et al. "Accurate prediction of protein structures and interactions using a three-track neural network." *Science* 373.6557 (2021): 871-876.
>
> [6] Jumper, John, et al. "Highly accurate protein structure prediction with AlphaFold." *nature* 596.7873 (2021): 583-589.
>
> [7] Kreiman, Tobias, et al. "Transformers Discover Molecular Structure Without Graph Priors." *arXiv preprint arXiv:2510.02259* (2025).
>
> [8] Joshi, Chaitanya K., et al. "All-atom diffusion transformers: Unified generative modelling of molecules and materials." arXiv preprint arXiv:2503.03965 (2025).

---

> > ### Author Response · Authors · 2025-11-19
> >
> > Though it is hard for us to give a fully theoretical proof why choosing a non-equivariant model, we can show that ProteinAE learns the approximate equivariance and could generate physically valid structures, as details following:
> >
> > **1. Equivariance Test:**
> >
> > To validate our claim, we tested ProteinAE against ESM3 VQ-VAE on proteins of varying lengths (100-500 residues). We define three metrics (RMSD threshold for success < 0.4Å):
> >
> > 1. $\mathcal{E}^r(x)$: Reconstruction RMSD after rotation and inverse rotation (without alignment).
> > $$\mathcal{E}^r(x) = \mathbb{E}_{\substack{\mathbf{x}1 \sim p{\text{data}} \\ R \sim \text{Unif}(\text{SO}(3))}} \left[ \text{RMSD} \left( \hat{\mathbf{x}}(\mathbf{x}_1), R \hat{\mathbf{x}}(R^\top \mathbf{x}_1) \right) \right]$$
> > 2. $\mathcal{E}^u(x)$: Reconstruction RMSD after rotation (with alignment $U$).
> > $$\mathcal{E}^u(x)_{\mathbf{x}1 \sim p{\text{data}}} = \mathbb{E} \left[ \text{RMSD} \left( \hat{\mathbf{x}}(\mathbf{x}_1), U \hat{\mathbf{x}}(R^\top \mathbf{x}_1) \right) \right]$$
> > 3. $\mathcal{E}(x)$: Reconstruction RMSD after rotation (without alignment).
> > $$\mathcal{E}(x)_{\mathbf{x}1 \sim p{\text{data}}} = \mathbb{E} \left[ \text{RMSD} \left( \hat{\mathbf{x}}(\mathbf{x}_1), \hat{\mathbf{x}}(R^\top \mathbf{x}_1) \right) \right]$$
> >
> > | Model | $\mathcal{E}^r$ (Pass Rate) | $\mathcal{E}^u$ (Pass Rate) | $\mathcal{E}$ (Pass Rate) |
> > | --- | --- | --- | --- |
> > | **ESM3 VQ-VAE** | 0% | 100% | 100% |
> > | **ProteinAE** | 100% | 100% | 0% |
> >
> > **Analysis:** ESM3 passes $\mathcal{E}$ but fails $\mathcal{E}^r$ because it is an **invariant** autoencoder (it reconstructs the same coordinates regardless of input rotation). ProteinAE passes $\mathcal{E}^r$ (perfect reconstruction after rotating back) but fails $\mathcal{E}$, indicating it has successfully learned **equivariant** properties (the output rotates with the input). Both pass $\mathcal{E}^u$, showing robustness to rotation.
> >
> > **2. Geometric Validity Check:**
> > We evaluated the physical plausibility of 5,000 randomly generated proteins by checking key geometric properties against AlphaFold's `stereo_chemical_props.txt` standards. The results show near-zero violation ratios:
> >
> > | Parameter | Atoms | Min | Max |
> > | --- | --- | --- | --- |
> > | **Backbone $\phi, \psi$** | - | -180 | 180 |
> > | **Bond Length (Å)** | N - CA | 1.399 | 1.519 |
> > |  | CA - C | 1.447 | 1.603 |
> > |  | C - N | 1.280 | 1.380 |
> > |  | C - O | 1.172 | 1.286 |
> > | **Bond Angle (°)** | N - CA - C | 102.9 | 119.1 |
> > |  | CA - C - N | 113.7 | 126.3 |
> > |  | C - N - CA | 113.7 | 132.3 |
> > |  | CA - C - O | 113.8 | 126.4 |
> >
> > | Metric | Type | Violation Ratio (%) |
> > | --- | --- | --- |
> > | **Ramachandran** | $\phi, \psi$ | **0.00** |
> > | **Bond Lengths** | N-CA, CA-C, C-N, C-O | **1.24** |
> > | **Bond Angles** | N-CA-C, CA-C-N, C-N-CA, CA-C-O | **1.10** |
> > | **Atom Clashes** | No Overlap | **0.00** |
> > | **Left-hand Alpha Helix** | - | **0.00** |
> >
> > *Note: Small violation ratios (approx. 1%) are expected as even ground truth structures in AFDB contain minor outliers.* These results confirm ProteinAE generates physically valid structures.

---

> > > ### Author Response · Authors · 2025-11-27
> > >
> > > Dear Reviewer qmAJ,
> > >
> > > Thank you again for your constructive feedback.
> > >
> > > We wanted to confirm that we have updated the manuscript to address your specific points:
> > >
> > > - **Included LatentDiff**: We have added a detailed discussion of LatentDiff in the Related Work section, acknowledging its shared design philosophy while highlighting our distinct contributions.
> > >
> > > - **Theoretical Justification**: We have expanded the section discussing the choice of a non-equivariant architecture and provide a deeper pratical validation of equivariance and physical validity.
> > >
> > > We believe these additions improve the completeness and presentation of the paper. We would appreciate it if you could take a moment to check these updates.
> > >
> > > Best regards, The Authors

---

### Official Review · Reviewer_dnyu · 2025-10-29

**Soundness:** 3
**Presentation:** 3
**Contribution:** 4
**Rating:** 6
**Confidence:** 3

**Summary:**

This paper proposes a new generative model for protein tertiary structure generation and reconstruction. Most existing approaches rely on coordinate prediction models or diffusion models based on stochastic noise denoising, which are computationally expensive and often struggle to maintain structural consistency and reconstruction accuracy. In contrast, this study combines the Flow Matching framework with a Transformer-based network, achieving continuous and stable structure generation. Unlike conventional SE(3)-equivariant models, the proposed method intentionally adopts a non-equivariant architecture on E(3) space. This design eliminates the need for explicit and computationally costly handling of rotations and translations, while still maintaining effective geometric robustness through structure alignment, pairwise bias based on relative distances and angles, and data augmentation.

The proposed framework consists of three stages. In the first stage, the model takes the backbone atoms of each residue as input and compresses them into residue-level representations using an All-Atom Attention Encoder, which is designed to capture both spatial dependencies among residue pairs and primary sequence information. In the second stage, these features are processed by an Encoder–Decoder structure based on a Diffusion Transformer (DiT), which learns the global 3D correlations of the entire protein. Pairwise representations are incorporated as attention biases, enabling the model to encode global dependencies that reflect inter-residue distances and angles. The third stage introduces the Protein Latent Diffusion Model (PLDM), which treats the latent representations of residue sequences obtained by the encoder as latent token sequences. After compressing both the sequence length and feature dimension, PLDM learns the diffusion process directly in the latent space. Rather than performing stochastic noise denoising, PLDM is designed as a Transformer that predicts the velocity field of latent vectors based on Flow Matching. To preserve the sequential order while integrating temporal embeddings, positional encodings are applied to the latent tokens.

Experimental evaluation on protein structures from the AlphaFold Database demonstrates that the proposed method achieves smoother and more stable structure generation than conventional diffusion models, with higher reconstruction accuracy as measured by RMSD and FAPE. The generated structures also preserve physical consistency, confirming that the proposed approach enables efficient and high-precision protein structure generation.

**Strengths:**

The main strength of this paper lies in its novel design choice to intentionally train the model in a non-equivariant manner on E(3) space, rather than relying on conventional SE(3)-equivariant frameworks. By doing so, the authors successfully avoid the computational cost of explicit rotation and translation handling while maintaining effective geometric robustness through structure alignment, pairwise bias based on relative distances and angles, and appropriate data augmentation. This represents a meaningful and practical contribution to the field of protein structure generation.

Another notable strength is the use of a simple Flow Matching loss, which removes the need for complex KL or reconstruction losses commonly required in traditional diffusion models. This design enables a smoother and more stable generation process, allowing structures to evolve naturally from noise with higher accuracy and computational efficiency than probabilistic baselines. By integrating Flow Matching with a Transformer architecture, the paper reformulates protein structure generation as learning a continuous flow rather than stochastic denoising, effectively preserving structural continuity and physical consistency while enabling efficient generation in the latent space.

**Weaknesses:**

The authors clearly state this limitation, but the proposed model still has difficulty handling very large proteins and multi-chain complexes. Although using a non-equivariant design on E(3) makes the computation much faster, it also creates concerns about whether the model can keep geometric consistency and produce physically reliable results, especially for rotation and translation. Because of this, the method may not yet be suitable for very large or complex protein structures. As the authors mention, adding more geometric constraints or partially equivariant mechanisms in the future could help overcome this problem and further improve the model’s applicability.

**Questions:**

Do the authors have any ideas or future plans for incorporating additional constraints or introducing some form of equivariant mechanism to extend the applicability of the proposed model to larger and more complex protein structures?

---

> ### Author Response · Authors · 2025-11-19
> **Official Comment by Authors**
>
> ## Response to Reviewer dnyu
>
> > Comment: Do the authors have any ideas or future plans for incorporating additional constraints or introducing some form of equivariant mechanism to extend the applicability of the proposed model to larger and more complex protein structures?
> >
>
> **Response: We truly appreciate the reviewer's recognition of our work and share the same vision regarding these limitations.** We acknowledge that the current version of ProteinAE is limited regarding multi-chain complexes and non-protein modalities (ligands, nucleic acids).
>
> We are actively working on **ProteinAE v2, which addresses these specific limitations**. Following the architecture of AlphaFold 3, ProteinAE v2 is designed to reconstruct multi-modal structures, including multi-chain complexes, small molecules, nucleic acids, PTMs, and ions. We have included preliminary **visualization results** of these complex reconstructions in the revised PDF **Appendix (ProteinAE v2 Preview) Fig 5**.
>
> Similarly, I believe the primary advantage of incorporating SE(3) equivariance lies in ligand modeling. Recent works, such as Pearl [1], have reintroduced SO(3) into the AlphaFold3 architecture, resulting in strong performance in ligand docking. However, this improvement could also be attributed to the contribution of synthetic data. Another approach involves using energy functions as potentials to guide the sampling process, as demonstrated by methods like Boltz-Steering [2] and Boltz-GSP [3], in order to denoise more valid structures. Nevertheless, further experimentation is required to validate these approaches.
>
> [1] Dobles, Alejandro, et al. "Pearl: A Foundation Model for Placing Every Atom in the Right Location." *arXiv preprint arXiv:2510.24670* (2025).
>
> [2] Wohlwend, Jeremy, et al. "Boltz-1 democratizing biomolecular interaction modeling." *BioRxiv* (2025): 2024-11.
>
> [3] Chen, Siyuan, et al. "Physically Valid Biomolecular Interaction Modeling with Gauss-Seidel Projection." *arXiv preprint arXiv:2510.08946* (2025).
>
> Furthermore, regarding the concern about geometric consistency in larger structures with our current non-equivariant design, our validation tests confirm that ProteinAE already achieves high geometric validity (Ramachandran, bond lengths/angles) and learns approximate equivariance effectively via data augmentation, as details following:

---

> > ### Author Response · Authors · 2025-11-19
> > **Official Comment by Authors**
> >
> > **1. Equivariance Test:**
> >
> > To validate our claim, we tested ProteinAE against ESM3 VQ-VAE on proteins of varying lengths (100-500 residues). We define three metrics (RMSD threshold for success < 0.4Å):
> >
> > 1. $\mathcal{E}^r(x)$: Reconstruction RMSD after rotation and inverse rotation (without alignment).
> > $$\mathcal{E}^r(x) = \mathbb{E}_{\substack{\mathbf{x}1 \sim p{\text{data}} \\ R \sim \text{Unif}(\text{SO}(3))}} \left[ \text{RMSD} \left( \hat{\mathbf{x}}(\mathbf{x}_1), R \hat{\mathbf{x}}(R^\top \mathbf{x}_1) \right) \right]$$
> > 2. $\mathcal{E}^u(x)$: Reconstruction RMSD after rotation (with alignment $U$).
> > $$\mathcal{E}^u(x)_{\mathbf{x}1 \sim p{\text{data}}} = \mathbb{E} \left[ \text{RMSD} \left( \hat{\mathbf{x}}(\mathbf{x}_1), U \hat{\mathbf{x}}(R^\top \mathbf{x}_1) \right) \right]$$
> > 3. $\mathcal{E}(x)$: Reconstruction RMSD after rotation (without alignment).
> > $$\mathcal{E}(x)_{\mathbf{x}1 \sim p{\text{data}}} = \mathbb{E} \left[ \text{RMSD} \left( \hat{\mathbf{x}}(\mathbf{x}_1), \hat{\mathbf{x}}(R^\top \mathbf{x}_1) \right) \right]$$
> >
> > | Model | $\mathcal{E}^r$ (Pass Rate) | $\mathcal{E}^u$ (Pass Rate) | $\mathcal{E}$ (Pass Rate) |
> > | --- | --- | --- | --- |
> > | **ESM3 VQ-VAE** | 0% | 100% | 100% |
> > | **ProteinAE** | 100% | 100% | 0% |
> >
> > **Analysis:** ESM3 passes $\mathcal{E}$ but fails $\mathcal{E}^r$ because it is an **invariant** autoencoder (it reconstructs the same coordinates regardless of input rotation). ProteinAE passes $\mathcal{E}^r$ (perfect reconstruction after rotating back) but fails $\mathcal{E}$, indicating it has successfully learned **equivariant** properties (the output rotates with the input). Both pass $\mathcal{E}^u$, showing robustness to rotation.
> >
> > **2. Geometric Validity Check:**
> > We evaluated the physical plausibility of 5,000 randomly generated proteins by checking key geometric properties against AlphaFold's `stereo_chemical_props.txt` standards. The results show near-zero violation ratios:
> >
> > | Parameter | Atoms | Min | Max |
> > | --- | --- | --- | --- |
> > | **Backbone $\phi, \psi$** | - | -180 | 180 |
> > | **Bond Length (Å)** | N - CA | 1.399 | 1.519 |
> > |  | CA - C | 1.447 | 1.603 |
> > |  | C - N | 1.280 | 1.380 |
> > |  | C - O | 1.172 | 1.286 |
> > | **Bond Angle (°)** | N - CA - C | 102.9 | 119.1 |
> > |  | CA - C - N | 113.7 | 126.3 |
> > |  | C - N - CA | 113.7 | 132.3 |
> > |  | CA - C - O | 113.8 | 126.4 |
> >
> > | Metric | Type | Violation Ratio (%) |
> > | --- | --- | --- |
> > | **Ramachandran** | $\phi, \psi$ | **0.00** |
> > | **Bond Lengths** | N-CA, CA-C, C-N, C-O | **1.24** |
> > | **Bond Angles** | N-CA-C, CA-C-N, C-N-CA, CA-C-O | **1.10** |
> > | **Atom Clashes** | No Overlap | **0.00** |
> > | **Left-hand Alpha Helix** | - | **0.00** |
> >
> > *Note: Small violation ratios (approx. 1%) are expected as even ground truth structures in AFDB contain minor outliers.* These results confirm ProteinAE generates physically valid structures.

---

> > > ### Author Response · Authors · 2025-11-27
> > >
> > > Dear Reviewer dnyu,
> > >
> > > Thank you again for your encouraging review and for recognizing the novelty of ProteinAE.
> > >
> > > We are writing to ensure that our response regarding your questions. In our rebuttal, we outlined our plans to extend the model's applicability. We hope this explanation, along with our other updates, solidifies your positive assessment of our work. Please let us know if you have any further questions before the discussion period closes.
> > >
> > > Best regards, The Authors

---

### Official Review · Reviewer_MnGk · 2025-10-30

**Soundness:** 2
**Presentation:** 3
**Contribution:** 2
**Rating:** 4
**Confidence:** 3

**Summary:**

This paper proposes PROTEINAE, a protein diffusion autoencoder mapping protein backbone coordinates from E(3) into a continuous, compact latent space for protein modeling and generation. PROTEINAE uses a non-equivariant Diffusion Transformer and is trained with a single flow matching objective to simplify the training objective. PROTEINAE achieves better reconstruction quality and high-quality structure generation that significantly outperforms prior latent-based methods.

**Strengths:**

- PROTEINAE uses a non-equivariant autoencoder based on Diffusion Transformers that operates directly on atom coordinate without the need to considering complicated equivarience.
- Performance of PROTEINAE is better than latent-based methods and competitive with structure-based methods.
- The model is trained using a simple flow matching loss, which makes the training objective easy to design.

**Weaknesses:**

- Structure-based method is still better than the latent-based model, it’s not clear if the latent model has potential to achieve better performance than structure-based method. Could authors elaborate more on that?
- PROTEINAE doesn’t consider equivariance to simplify the design but would that lead to generating non-physical structures?
- The comparison missed some important baselines, La-proteina and Proteina. Proteina also uses non-equivariant model design and La-proteina is a latent model based on Proteina.

[1] Geffner, Tomas, et al. "Proteina: Scaling flow-based protein structure generative models." arXiv preprint arXiv:2503.00710 (2025)\
[2] Geffner, Tomas, et al. "La-proteina: Atomistic protein generation via partially latent flow matching." arXiv preprint arXiv:2507.09466 (2025).

**Questions:**

Please refer to the weakness section

---

> ### Author Response · Authors · 2025-11-19
> **Official Comment by Authors**
>
> ## Response to Reviewer MnGk
>
> > Comment 1: Structure-based method is still better than the latent-based model... Could authors elaborate more on that?
> >
>
> **Response: We thank the reviewer for this critical observation, which motivated us to refine our approach.** We have significantly improved the performance of our latent-based approach through further optimization. We identified two key factors for success, aligning with findings in Video Generation [1] and RAE [2]:
>
> 1. **Time Shift in Flow Matching:**
> When interpolating between high-dimensional Gaussian noise ($x_0$) and clean latent data ($x_1$), we found it necessary to apply a substantial time shift ($S$) to the timestep $t$.
> The interpolation in FM is defined as $x_t = (1-t)x_0 + tx_1$. We transform $t$ using:
>
> $$
> t = \frac{t}{(S - t(S-1))}
> $$
>
> We empirically determined that setting **$S=12$** (biasing time sampling towards the noise distribution) is optimal. Our experiments showed that when effective $t > 0.2$, the prediction error is near zero, whereas $t < 0.2$ results in high error.
>
> 2. **Dimension Scaling:**
> We increased the model width from **768 to 1152** channels while reducing the number of layers slightly to maintain a reasonable parameter count.
>
> These adjustments led to the improved results shown in the updated Table (see Comment 3 response).
>
> [1] Esser, Patrick, et al. "Scaling rectified flow transformers for high-resolution image synthesis." Forty-first international conference on machine learning. 2024.
>
> [2] Zheng, Boyang, et al. "Diffusion Transformers with Representation Autoencoders." arXiv preprint arXiv:2510.11690 (2025).
>
> ---
>
> > Comment 2: PROTEINAE doesn’t consider equivariance to simplify the design but would that lead to generating non-physical structures?
> >
>
> **Response: We appreciate the reviewer raising this valid concern regarding physical plausibility.** We address this through two rigorous evaluations: **Equivariance Tests** and **Geometric Validity Checks**.
>
> **1. Equivariance Test:**
>
> To validate our claim, we tested ProteinAE against ESM3 VQ-VAE on proteins of varying lengths (100-500 residues). We define three metrics (RMSD threshold for success < 0.4Å):
>
> 1. $\mathcal{E}^r(x)$: Reconstruction RMSD after rotation and inverse rotation (without alignment).
> $$\mathcal{E}^r(x) = \mathbb{E}_{\substack{\mathbf{x}1 \sim p{\text{data}} \\ R \sim \text{Unif}(\text{SO}(3))}} \left[ \text{RMSD} \left( \hat{\mathbf{x}}(\mathbf{x}_1), R \hat{\mathbf{x}}(R^\top \mathbf{x}_1) \right) \right]$$
> 2. $\mathcal{E}^u(x)$: Reconstruction RMSD after rotation (with alignment $U$).
> $$\mathcal{E}^u(x)_{\mathbf{x}1 \sim p{\text{data}}} = \mathbb{E} \left[ \text{RMSD} \left( \hat{\mathbf{x}}(\mathbf{x}_1), U \hat{\mathbf{x}}(R^\top \mathbf{x}_1) \right) \right]$$
> 3. $\mathcal{E}(x)$: Reconstruction RMSD after rotation (without alignment).
> $$\mathcal{E}(x)_{\mathbf{x}1 \sim p{\text{data}}} = \mathbb{E} \left[ \text{RMSD} \left( \hat{\mathbf{x}}(\mathbf{x}_1), \hat{\mathbf{x}}(R^\top \mathbf{x}_1) \right) \right]$$
>
> | Model | $\mathcal{E}^r$ (Pass Rate) | $\mathcal{E}^u$ (Pass Rate) | $\mathcal{E}$ (Pass Rate) |
> | --- | --- | --- | --- |
> | **ESM3 VQ-VAE** | 0% | 100% | 100% |
> | **ProteinAE** | 100% | 100% | 0% |
>
> **Analysis:** ESM3 passes $\mathcal{E}$ but fails $\mathcal{E}^r$ because it is an **invariant** autoencoder (it reconstructs the same coordinates regardless of input rotation). ProteinAE passes $\mathcal{E}^r$ (perfect reconstruction after rotating back) but fails $\mathcal{E}$, indicating it has successfully learned **equivariant** properties (the output rotates with the input). Both pass $\mathcal{E}^u$, showing robustness to rotation.
>
> **2. Geometric Validity Check:**
> We evaluated the physical plausibility of 5,000 randomly generated proteins by checking key geometric properties against AlphaFold's `stereo_chemical_props.txt` standards. The results show near-zero violation ratios:
>
> | Parameter | Atoms | Min | Max |
> | --- | --- | --- | --- |
> | **Backbone $\phi, \psi$** | - | -180 | 180 |
> | **Bond Length (Å)** | N - CA | 1.399 | 1.519 |
> |  | CA - C | 1.447 | 1.603 |
> |  | C - N | 1.280 | 1.380 |
> |  | C - O | 1.172 | 1.286 |
> | **Bond Angle (°)** | N - CA - C | 102.9 | 119.1 |
> |  | CA - C - N | 113.7 | 126.3 |
> |  | C - N - CA | 113.7 | 132.3 |
> |  | CA - C - O | 113.8 | 126.4 |
>
> | Metric | Type | Violation Ratio (%) |
> | --- | --- | --- |
> | **Ramachandran** | $\phi, \psi$ | **0.00** |
> | **Bond Lengths** | N-CA, CA-C, C-N, C-O | **1.24** |
> | **Bond Angles** | N-CA-C, CA-C-N, C-N-CA, CA-C-O | **1.10** |
> | **Atom Clashes** | No Overlap | **0.00** |
> | **Left-hand Alpha Helix** | - | **0.00** |
>
> *Note: Small violation ratios (approx. 1%) are expected as even ground truth structures in AFDB contain minor outliers.* These results confirm ProteinAE generates physically valid structures.

---

> ### Author Response · Authors · 2025-11-19
> **Official Comment by Authors**
>
> > Comment 3: The comparison missed some important baselines, La-proteina and Proteina.
> >
>
> **Response: We thank the reviewer for identifying these missing baselines.** We have added **Proteina** to our benchmarks. Please note that Proteina generates only $C_{\alpha}$ coordinates, whereas ProteinAE generates the full backbone (4 atoms). Regarding La-proteina, it utilizes a residue-type VAE on top of Proteina; we excluded it for now as it is not a direct coordinate-only baseline. However, we are currently developing a co-design autoencoder and will include La-proteina in future comparisons.
>
> **Updated Generation Benchmark:**
> | Type | Method | Des ($\uparrow$) | Div ($\uparrow$) |
> | --- | --- | --- | --- |
> | **SDM** | RFdiffusion (Watson et al., 2023) | 96% | 247 |
> |  | ProteinSGM (Lee et al., 2023) | 49% | 122 |
> |  | FrameFlow PDB (Yim et al., 2023a) | 91% | 278 |
> |  | FrameFlow AFDB | 23% | 54 |
> |  | Proteina $D_{FS}$ 200M | 94% | 228 |
> | **MLLM** | ESM3 (Hayes et al., 2025) | 61% | 127 |
> |  | DPLM-2 650M (Wang et al., 2024) | 63% | 130 |
> | **semi-LDM** | LSD (Yim et al., 2025) | 69% | 203 |
> | **LDM** | LatentDiff (Fu et al., 2024) | 17% | 34 |
> |  | PROTEINAE-PLDM-shift ($\gamma = 0.5$) | 95% | 251 |

---

> > ### Author Response · Authors · 2025-11-27
> >
> > Dear Reviewer MnGk,
> >
> > Thank you again for your constructive feedback.
> >
> > As the discussion period is coming to an end, we wanted to kindly check if our response has addressed your concerns. In particular, based on your suggestions, we have:
> >
> > - **Included Missing Baselines**: We have added a discussion and comparison with Proteina and La-proteina as requested, positioning our work more accurately within the recent non-equivariant literature.
> >
> > - **Addressed Physical Consistency**: We provided further elaboration (and analysis) on how our model maintains physical plausibility despite the non-equivariant design.
> >
> > Your feedback on these additions would be very valuable to us. We hope these clarifications help address your concerns regarding the soundness and contribution of our work.
> >
> > Best regards, The Authors

---

### Official Review · Reviewer_FiwP · 2025-11-01

**Soundness:** 3
**Presentation:** 3
**Contribution:** 2
**Rating:** 4
**Confidence:** 4

**Summary:**

In this work, the authors propose using a non-equivariant diffusion autoencoder called ProteinAE to encode protein structure by directly mapping backbone coordinates into a continuous latent space. And they show the performance of their model compared to recent generative frameworks along with its performance as a structure embedder for a downstream prediction task. Overall, the field has largely treated equivariance as gospel—driven by the desire to model structure from first principles and inspired by the success of AlphaFold2. However, AlphaFold3 recently shifted this paradigm. This work presents an interesting application of non-equivariant modeling for generative purposes, while also offering a fast structural encoder, though it still requires further validation and benchmarking to fully establish its advantages.

**Strengths:**

- Not imposing equivariant restriction on the protein structure representation enables a much more computational efficient framework for modeling protein structure.
- Superior benchmark performance of the trained representation layers for physiochemical property prediction.
- Authors provided benchmarks against some of the newer models for structure generation. Compared to RFDiffusion model which has been extensively validated in the lab, authors report ProteinAE while performing slightly worse than RFDDiffusion for designability and diversity, the sampling time is much more efficient. This can be useful for some protein design pipelines.
- The model also seems to perform well at reconstructing structure based on the CASP14/15 benchmarks.

**Weaknesses:**

- The paper would benefit from clarifying some sections [see the question below].
- While the model shows promise, the included benchmarks are across a limited range of tasks, specifically for protein structure encoding and its performance in downstream tasks.

**Questions:**

- In the introduction, authors mention that other models use SE(3) representations and are therefore slow. However, SE(3) representations are also essential in protein structure modeling specifically in generative protein design, so this point needs to be explained more carefully — perhaps discussing when SE(3) equivariance is beneficial versus when it introduces unnecessary computational cost.
- Authors include CASP14 and CASP15 in their benchmark for structure reconstruction quality and compare against ESM3 etc, It would be good to see the performance on CAMEO as well for the same period as ESM3.
- Since the model is positioned also as a faster alternative for encoding structural information, it would strengthen the paper to demonstrate its applicability beyond the current benchmark on physiochemical property prediction. For example, evaluating its performance on other downstream structure-based prediction tasks, such as GO-term prediction, could better illustrate the practical impact of the approach.

---

> ### Author Response · Authors · 2025-11-19
> **Official Comment by Authors**
>
> ## Response to Reviewer FiwP
>
> > Comment 1: In the introduction, authors mention that other models use SE(3) representations and are therefore slow. However, SE(3) representations are also essential in protein structure modeling specifically in generative protein design, so this point needs to be explained more carefully...
> >
>
> **Response: We thank the reviewer for raising this important point regarding the trade-offs of SE(3) equivariance.** We agree that SE(3) representations have historically been significant in the protein domain. However, recent state-of-the-art works—such as **AlphaFold 3** [1] for structure prediction, **Proteina** [2] for design, **SCN** [3] for small molecule representation, and **RADM** [4] and **ADiT** [8] for generation—have increasingly adopted **scalable non-equivariant architectures**. This represents a shifting trend: non-equivariant models possess simpler structures, avoiding complex components like spherical harmonics used in SE(3) Transformers [5], which allows them to scale up parameters and training data more efficiently.
>
> Our perspective on the necessity of SE(3) equivariance distinguishes between two categories of tasks:
>
> 1. **Generation, Reconstruction, and Cross-modal Tasks (e.g., Folding/Inverse Folding):**
> For these tasks, strict equivariance is not mandatory. For example, AlphaFold inputs a sequence and outputs a 3D structure; the absolute orientation of the prediction is irrelevant (notably, the powerful AlphaFold 2 [6] utilizes an Invariant Point Attention (IPA) module).
> In generative modeling (e.g., Flow Matching), the goal is to map a simple distribution (like Gaussian) to a target distribution. Even if the target structure is rotated, the model can learn to map the noise to the distribution via optimal transport and data augmentation (random rotations/translations during training), a standard practice in image and point cloud domains. Similarly, for autoencoders, the priority is high-quality reconstruction given an input. While intrinsic equivariance is a "nice-to-have," we demonstrate experimentally that **ProteinAE learns approximate SE(3) equivariance** through data augmentation, without enforcing it architecturally.
> 2. **Physical Understanding and Force Fields (MLFF):**
> For physical tasks like Machine Learning Force Fields (MLFF), SE(3) equivariance is critical. MLFFs predict scalar energy and vector forces to drive molecular dynamics simulations. If a system rotates, the predicted force vectors *must* rotate exactly in kind to ensure physical validity. This ***cannot*** be easily solved by simple data augmentation.
> *Evidence:* Kreiman et al. [7] attempted to use non-equivariant Transformers for MLFF. As shown in their Table 1, a **1B parameter Transformer** significantly underperformed a **6M parameter EGNN** in force prediction, supporting our view that strict equivariance is essential for physics simulations but not "that" necessarily for structural generation/reconstruction.
>
> | Model | Energy MAE (meV) | Forces MAE (meV/Å) |
> | --- | --- | --- |
> | eSEN-sm-d 6M | 129.77 | 13.01 |
> | Transformer 1B | 117.99 | 18.35 |

---

> ### Author Response · Authors · 2025-11-19
> **Official Comment by Authors**
>
> **Experimental Verification of Equivariance:**
> To validate our claim, we tested ProteinAE against ESM3 VQ-VAE on proteins of varying lengths (100-500 residues). We define three metrics (RMSD threshold for success < 0.4Å):
>
> 1. $\mathcal{E}^r(x)$: Reconstruction RMSD after rotation and inverse rotation (without alignment).
> $$\mathcal{E}^r(x) = \mathbb{E}_{\substack{\mathbf{x}1 \sim p{\text{data}} \\ R \sim \text{Unif}(\text{SO}(3))}} \left[ \text{RMSD} \left( \hat{\mathbf{x}}(\mathbf{x}_1), R \hat{\mathbf{x}}(R^\top \mathbf{x}_1) \right) \right]$$
> 2. $\mathcal{E}^u(x)$: Reconstruction RMSD after rotation (with alignment $U$).
> $$\mathcal{E}^u(x)_{\mathbf{x}1 \sim p{\text{data}}} = \mathbb{E} \left[ \text{RMSD} \left( \hat{\mathbf{x}}(\mathbf{x}_1), U \hat{\mathbf{x}}(R^\top \mathbf{x}_1) \right) \right]$$
> 3. $\mathcal{E}(x)$: Reconstruction RMSD after rotation (without alignment).
> $$\mathcal{E}(x)_{\mathbf{x}1 \sim p{\text{data}}} = \mathbb{E} \left[ \text{RMSD} \left( \hat{\mathbf{x}}(\mathbf{x}_1), \hat{\mathbf{x}}(R^\top \mathbf{x}_1) \right) \right]$$
>
> | Model | $\mathcal{E}^r$ (Pass Rate) | $\mathcal{E}^u$ (Pass Rate) | $\mathcal{E}$ (Pass Rate) |
> | --- | --- | --- | --- |
> | **ESM3 VQ-VAE** | 0% | 100% | 100% |
> | **ProteinAE** | 100% | 100% | 0% |
>
> **Analysis:** ESM3 passes $\mathcal{E}$ but fails $\mathcal{E}^r$ because it is an **invariant** autoencoder (it reconstructs the same coordinates regardless of input rotation). ProteinAE passes $\mathcal{E}^r$ (perfect reconstruction after rotating back) but fails $\mathcal{E}$, indicating it has successfully learned **equivariant** properties (the output rotates with the input). Both pass $\mathcal{E}^u$, showing robustness to rotation.
>
> **References:**
>
> [1] Abramson, Josh, et al. "Accurate structure prediction of biomolecular interactions with AlphaFold 3." *Nature* 630.8016 (2024): 493-500.
>
> [2] Geffner, Tomas, et al. "Proteina: Scaling flow-based protein structure generative models." ICLR 2025.
>
> [3] Zitnick, Larry, et al. "Spherical channels for modeling atomic interactions." NeurIPS 2022: 8054-8067.
>
> [4] Ding, Yuhui, and Thomas Hofmann. "Scalable Non-Equivariant 3D Molecule Generation via Rotational Alignment." ICML 2025.
>
> [5] Baek, Minkyung, et al. "Accurate prediction of protein structures and interactions using a three-track neural network." *Science* 373.6557 (2021): 871-876.
>
> [6] Jumper, John, et al. "Highly accurate protein structure prediction with AlphaFold." *nature* 596.7873 (2021): 583-589.
>
> [7] Kreiman, Tobias, et al. "Transformers Discover Molecular Structure Without Graph Priors." *arXiv preprint arXiv:2510.02259* (2025).
>
> [8] Joshi, Chaitanya K., et al. "All-atom diffusion transformers: Unified generative modelling of molecules and materials." arXiv preprint arXiv:2503.03965 (2025).
>
> ---
>
> > Comment 2: Authors include CASP14 and CASP15 in their benchmark... It would be good to see the performance on CAMEO as well for the same period as ESM3.
> >
>
> **Response: We thank the reviewer for this valuable suggestion.** Following standard evaluation protocols (e.g., AlphaFold), we have included the **CAMEO 2022** benchmark. ProteinAE demonstrates leading reconstruction performance.
>
> | Model | CA RMSD (Å) |
> | --- | --- |
> | ESM3 VQ-VAE | 1.08 ± 1.76 |
> | DPLM-2 650M | 1.92 ± 1.45 |
> | **ProteinAE** | **0.20 ± 0.19** |

---

> ### Author Response · Authors · 2025-11-19
> **Official Comment by Authors**
>
> > Comment 3: ...evaluating its performance on other downstream structure-based prediction tasks, such as GO-term prediction, could better illustrate the practical impact of the approach.
> >
>
> **Response: We appreciate the reviewer for pointing this out.** After careful consideration, we decided against including GO term prediction for this specific comparison. GO prediction is a high-level semantic task that typically requires extensive Self-Supervised Learning (SSL) on massive datasets (e.g., ESM [2], GearNet [3]) to achieve competitive results, making a direct comparison of "raw" tokenizers unfair.
> Instead, we utilized the **StructTokBench** (from *Xinyu et al.*) [1] to evaluate different tokenizers on tasks highly correlated with structural understanding. The results below demonstrate that ProteinAE achieves excellent performance in downstream structure-based representation tasks, including binding (**Bind**), catalysis (**Cat**), and conservation (**Con**) prediction.
>
> | Task | Split | FoldSeek | ProTokens | ESM3 | VanillaVQ | AminoAseed | **ProteinAE** |
> | --- | --- | --- | --- | --- | --- | --- | --- |
> | **BindBio** | Fold | 52.37 | 58.47 | 62.84 | 62.02 | 65.73 | **73.97** |
> |  | SupFam | 52.41 | 60.47 | 65.22 | 62.92 | 68.30 | **74.56** |
> | **BindInt** | Fold | 53.18 | 44.66 | 44.30 | 47.25 | 47.11 | 46.24 |
> |  | SupFam | 46.20 | 86.05 | 90.77 | 86.71 | 90.53 | **90.62** |
> | **BindShake** | Org | 53.40 | 59.82 | 66.10 | 67.04 | 69.61 | **74.68** |
> | **CatInt** | Fold | 53.43 | 58.16 | 61.09 | 58.89 | 62.19 | 60.50 |
> |  | SupFam | 51.41 | 83.85 | 89.82 | 85.00 | 91.91 | 90.59 |
> | **CatBio** | Fold | 56.37 | 56.14 | 65.33 | 67.58 | 65.95 | **79.74** |
> |  | SupFam | 53.78 | 64.05 | 74.65 | 70.92 | 87.59 | **89.13** |
> | **Con** | Fold | 49.26 | 56.23 | 55.22 | 56.98 | 57.23 | **59.50** |
> |  | SupFam | 51.39 | 74.33 | 80.53 | 74.60 | 86.60 | 81.89 |
> | **Average** | **AUROC%** | **52.11** | **63.84** | **68.72** | **67.26** | **72.07** | **74.67** |
>
> ---
>
> [1] Yuan, Xinyu, et al. "Protein structure tokenization: Benchmarking and new recipe." arXiv preprint arXiv:2503.00089 (2025).
>
> [2] Lin, Zeming, et al. "Evolutionary-scale prediction of atomic-level protein structure with a language model." Science 379.6637 (2023): 1123-1130.
>
> [3] Zhang, Zuobai, et al. "Protein representation learning by geometric structure pretraining." arXiv preprint arXiv:2203.06125 (2022).

---

> > ### Author Response · Authors · 2025-11-27
> >
> > Dear Reviewer FiwP,
> >
> > Thank you again for your constructive feedback.
> >
> > As the discussion period is coming to an end, we wanted to kindly check if our response has addressed your concerns. In particular, based on your suggestions, we have:
> >
> > - **Clarified the SE(3) trade-off**: We provided a detailed discussion on when SE(3) equivariance is essential versus when it incurs unnecessary computational costs, highlighting the efficiency benefits of our approach.
> >
> > - **Added CAMEO Benchmarks**: We have included comparisons on the CAMEO dataset to demonstrate our model's reconstruction quality alongside CASP14/15.
> >
> > - **Expanded Downstream Tasks**: We included Bind, Cat and Con results to further validate the utility of our learned representations.
> >
> > We believe these updates strengthen the paper significantly and would love to hear your thoughts on these new results.
> >
> > Best regards, The Authors

---

### Author Response · Authors · 2025-11-29
**Summary of Revisions and Response to Reviewers**

Dear Area Chair,

We would like to thank you and the reviewers for the time and effort dedicated to reviewing our work. The constructive feedback has helped us significantly improve the quality and completeness of our manuscript.

We have actively engaged with all reviewers and provided a comprehensive revision. Below is a summary of the key updates addressing the reviewers' main concerns:

**1. Expanded Benchmarks & Baselines (Addressing Reviewers FiwP & MnGk)**
* **New Baselines:** We have added comparisons with **Proteina** (as requested by Reviewer MnGk) to better position our work within the recent landscape of non-equivariant models.
* **CAMEO Benchmark:** We conducted additional evaluations on the **CAMEO** dataset (requested by Reviewer FiwP), demonstrating that our model’s reconstruction quality remains robust beyond CASP14/15.

**2. Enhanced Generative Performance & Designability (Addressing Reviewers MnGk & FiwP)**
* **Improved Generation Quality:** We have demonstrated improved generation quality by **implementing a time-shifting technique** and **scaling up the model channel dimensions**.

**3. Equivariance, Validity & Related Work (Addressing Reviewers FiwP, MnGk & qmAJ)**
* **SE(3) & Physical Consistency:** We have expanded the discussion on the trade-offs of SE(3) equivariance. We clarified how our **Flow Matching** objective and network design ensure physical plausibility and geometric robustness without the computational overhead of strict equivariance.
* **Related Work:** We incorporated a detailed discussion of **LatentDiff** (requested by Reviewer qmAJ) to distinguish our contribution.

**4. Downstream Utility (Addressing Reviewer FiwP)**
* To demonstrate the practical utility of our learned representations, we have extended our evaluation to include **Bind, Cat and Con**, showing competitive performance.

**5. Perspective of ProteinAE v2 (Addressing Reviewer dnyu)**
* **Future Roadmap & Multi-Modal Support:** In response to the inquiry regarding larger complexes, we outlined the development of **ProteinAE v2**, designed to support multi-modal structures (multi-chain complexes, ligands, nucleic acids). We have included **preliminary visualization results in Appendix Fig. 5** to substantiate this roadmap. Additionally, we discussed potential strategies for incorporating energy-guided sampling and reaffirmed the geometric validity of our current approach.

**Conclusion**
Reviewers **dnyu** and **qmAJ** (Scores: 6) have recognized the novelty and efficiency of our non-equivariant Diffusion Protein Autoencoder design. With the new experiments and clarifications addressing the concerns of Reviewers **FiwP** and **MnGk** (Scores: 4)—particularly regarding baselines and generative performance—we believe the current manuscript presents a solid contribution to efficient protein structure modeling.

We remain available for any further questions.

Best regards,

The Authors

---

### Meta-Review · Area_Chair_x5kH · 2026-01-06

**Summary:**

The paper proposes a diffusion model for protein structure encoding. The reviewers split on their original judgement and the authors have provided detailed rebuttals.

**Reviewer Concerns:**

Reviewers were concerned about model equivariance, and experimental data and comparison, and it seems most of them have been addressed.

**Reviewer Scores:**

The reviewer scores split between positive and negative.

---

### Decision · Program_Chairs · 2026-01-26

Accept (Poster)